# Maximum Independent Set:
# Self-Training through Dynamic Programming

**Lorenzo Brusca**[*]    **Lars C.P.M. Quaedvlieg**[*]    **Stratis Skoulakis**[*]
**Grigorios G Chrysos**[†]    **Volkan Cevher**
École Polytechnique Fédérale de Lausanne, Switzerland
{[first name].[last name]}@epfl.ch

## Abstract

This work presents a graph neural network (GNN) framework for solving the maximum independent set (MIS) problem, inspired by dynamic programming (DP). Specifically, given a graph, we propose a DP-like recursive algorithm based on GNNs that firstly constructs two smaller sub-graphs, predicts the one with the larger MIS, and then uses it in the next recursive call. To train our algorithm, we require annotated comparisons of different graphs concerning their MIS size. Annotating the comparisons with the output of our algorithm leads to a self-training process that results in more accurate self-annotation of the comparisons and vice versa. We provide numerical evidence showing the superiority of our method vs prior methods in multiple synthetic and real-world datasets.

## 1   Introduction

Deep neural networks (DNNs) have achieved unprecedented success in extracting intricate patterns directly from data without the need for handcrafted rules, while still generalizing well to new and previously unseen instances [He et al., 2016, Vaswani et al., 2017]. Among other applications, this success has led to the development of frameworks that utilize DNNs to solve combinatorial optimization (CO) problems, such as the Traveling Salesman Problem [Xing and Tu, 2020, Hu et al., 2020, Prates et al., 2019], the Job-Shop Scheduling Problem [Zhang et al., 2020, Park et al., 2021], and the Quadratic Assignment Problem [Nowak et al., 2017].

A core challenge for deep learning approaches on CO is the lack of training data. Annotating such data requires the solution of a huge number of instances of the CO, hence such supervised learning approaches are computationally infeasible for NP-hard problems [Yehuda et al., 2020]. Circumventing this difficulty is key to unlocking the full potential of otherwise broadly applicable DNNs for CO.

Our work demonstrates how classical ideas in CO together with DNNs can lead to a scalable self-supervised learning approach, mitigating the lack of training data. Concretely, we focus on the Maximum Independent Set (MIS) problem: Given a graph $G(V, E)$, MIS asks for a set of nodes of maximum cardinality such that no two nodes in the selected set are connected with an edge. MIS is an NP-hard problem with several hand-crafted heuristics (e.g., *greedy heuristic*, *local search*). More recently, several deep learning approaches have been proposed [Karalias and Loukas, 2020, Toenshoff et al., 2019, Schuetz et al., 2022a].

Our approach involves the following steps to determine an MIS in a graph. We use graph neural networks (GNNs) [Wu et al., 2020] to enable a model to generate approximate maximum independent sets after training on data that was annotated by the model itself. For this purpose, we draw inspiration

---

[*]These authors contributed equally to this work.

[†]Work done while at EPFL. Currently at University of Wisconsin-Madison; [surname]@wisc.edu.

37th Conference on Neural Information Processing Systems (NeurIPS 2023).

from dynamic programming (DP) and employ a DP-like recursive algorithm. Initially, we are given a graph. At each recursive step, we select a random vertex from that graph and create two sub-graphs: one by removing the selected vertex and another by removing all its neighboring vertices. We then make a comparison between these sub-graphs to determine which sub-graph is likely to have a larger independent set, and we use the sub-graph with the highest estimated independent set for the next recursive call. We repeat this process until we reach a graph consisting only of isolated vertices, which signifies the discovery of an independent set for the original graph.

Dynamic programming guarantees that if our predictions are accurate (i.e., we select the sub-graph with the largest MIS value), our recursive algorithm will always result in a maximum independent set. To make accurate predictions, we introduce "graph comparing functions," which take two graphs as input and output a winner. We implement such graph-comparing functions with GNNs.

We adopt a self-training approach to train our graph-comparing function and optimize the parameters of the GNN. In each epoch, we update the graph-comparing function parameters to ensure it accurately fits the data it has seen so far. The data comprises pairs of graphs $(G, G')$ along with a label $\text{Label}(G, G') \in \{0, 1\}$. For annotating the labels, we utilize the output of the recursive algorithm that leverages the graph-comparing function. Supported by theoretical and experimental evidence, we demonstrate how the self-annotation process improves parameter selection.

We conduct a thorough validation of our self-training approach in three real-world graph distribution datasets. Our algorithm surpasses the performance of previous deep learning methods [Karalias and Loukas, 2020, Toenshoff et al., 2019, Ahn et al., 2020] in the context of the MIS problem. To further validate the efficacy of our method, we explore its robustness on out-of-distribution data. Notably, our results demonstrate that the induced algorithm achieves competitive performance, showcasing the generalization capability of the learned comparator across different graph structures and distributions. In addition, we extend the evaluation of our DP-based self-training approach to tackle the Minimum Vertex Cover (MVC) problem in Appendix E. Encouragingly, similar to the MIS case, our induced GNN-based algorithms for MVC admit competitive performance with respect to other deep-learning approaches.

The code for the experiments and models discussed in this paper is available at: `https://github.com/LIONS-EPFL/dynamic-MIS`.

## 2   Related Work

Our work lies in the intersection of various domains, i.e., combinatorial optimization, Dynamic Programming, and (graph) neural networks. We review the most critical ideas in each domain here and defer a more detailed discussion in Appendix A.

**Graph Neural Networks (GNNs)** have gained widespread popularity due to their ability to learn representations of graph-structured data [Xiao et al., 2022, Zhang and Chen, 2018, Zhu et al., 2021, Errica et al., 2019] invariant to the size of the graph. More complex architectural blocks, such as the Graph Convolutional Network (GCN) [Kipf and Welling, 2017, Zhang et al., 2019], the Graph Attention Network (GAT) [Veličković et al., 2017], and the Graph Isomorphism Network [Xu et al., 2018] have become influential instances of GNNs. In our work, we utilize a simple GNN architecture to showcase the effectiveness of our proposed framework. While our choice of architecture is intentionally simple, we emphasize its modular nature, which enables us to incorporate more complex GNNs with ease.

**Combinatorial Optimization**: Supervised learning approaches have been used for tackling CO tasks, such as the Traveling Salesman Problem (TSP) [Vinyals et al., 2015], the Vehicle Routing Problem (VRP) [Shalaby et al., 2021], and Graph Coloring [Lemos et al., 2019]. Due to the graph structure of the problems, GNNs are often used for tackling those tasks [Prates et al., 2019, Nazari et al., 2018, Schuetz et al., 2022b]. However, owing to the computational overhead of obtaining supervised labels, such supervised approaches often do not scale well. Instead, unsupervised approaches have been deployed recently [Wang and Li, 2023]. A popular approach relies on a continuous relaxation of the loss function [Karalias and Loukas, 2020, Wang et al., 2022, Wang and Li, 2023]. In contrast to the previous unsupervised works, we adopt Dynamic Programming techniques to diminish the overall time complexity of the algorithm. Another approach uses reinforcement learning (RL) methods to address CO tasks, such as in Covering Salesman Problem [Li et al., 2021], the TSP [Zhang et al.,

2022], the VRP [James et al., 2019], and the Minimum Vertex Cover (MVC) [Tian and Li, 2021]. However, applying RL to CO problems can be challenging because of the long learning time required and the non-differentiable nature of the loss function.

**Dynamic Programming** has been a powerful problem-solving technique since at least the 50s [Bellman, 1954]. In recent years, researchers have explored the use of deep neural networks (DNNs) to replace the function responsible for dividing a problem into subproblems and estimating the optimal decision at each step [Yang et al., 2018]. Despite the progress, implementing CO tasks with Dynamic Programming suffers from significant computational overheads, since the size of the search space grows exponentially with the problem size [Xu et al., 2020]. Our approach overcomes this issue by utilizing a model that approximates the standard lookup table from Dynamic Programming, meaning that we avoid the exponential search space typically associated with DP.

## 3 An Optimal Solution to Maximum Independent Set (MIS)

Firstly, let us introduce MIS and its relationship with Dynamic Programming.

**Notation**: $G(V, E)$ denotes an undirected graph where $V$ stands for vertices and $E$ for the edges. $\mathcal{N}(v)$ denotes the neighbors of vertex $v \in V$, $\mathcal{N}(v) = \{u \in V \ : \ (u, v) \in E\}$. The *degree* of vertex $v \in V$ is $d(v) := |\mathcal{N}(v)|$. Given a set of vertices $S \subseteq V$, $G/S$ denotes the remaining graph of $G$ after removing all nodes $v \in S$.

**Definition 1** (Maximum Independent Set). *Given an undirected graph $G(V, E)$, find a maximum set of nodes $S \subseteq V$ such that $(u, v) \notin E$ for all vertices $u, v \in S$. We denote with $|\mathrm{MIS}(G)|$ the size of the maximum independent set of graph $G$.*

Dynamic Programming is a powerful technique for algorithmic design in which the optimal solution of the instance of interest is constructed by combining the optimal solution of smaller sub-instances. The combination step is governed by local optimality conditions which, in the context of MIS, take the form of Theorem 1. Theorem 1 establishes that the decision to remove a node $v \in V$ or its neighbors $\mathcal{N}(v)$ during the recursive process depends on whether $|\mathrm{MIS}\,(G/\mathcal{N}(v))| \geq |\mathrm{MIS}(G/v)|$. This decision continues until a graph with no edges is reached. According to Theorem 1, if at each step of the recursion, the choice is made based on whether $|\mathrm{MIS}\,(G/\mathcal{N}(v))| \geq |\mathrm{MIS}(G/v)\,|$ or not, then the resulting empty graph is guaranteed to be an optimal solution. The proof is this theorem can be found in Appendix I.

**Theorem 1.** *Let a graph $G(V, E) \in \mathcal{G}$. Then for any vertex $v \in V$ with $d(v) \geq 1$,*

$$|\mathrm{MIS}(G)| = \max\left(|\mathrm{MIS}\,(G/\mathcal{N}(v))|, |\mathrm{MIS}(G/\{v\})\,|\right)^1 .$$

## 4 Graph Neural Network-Based Algorithm for MIS

In this section, we present our approach for developing algorithms for MIS parameterized by parameters $\theta \in \Theta$. Initially in Sec. 4.1 we present how any *graph-comparing function* taking as input two different graphs and outputting a $\{0, 1\}$ value can be used in the construction of an algorithm computing an independent set (not necessarily optimal). In Sec. 4.2 we present how Graph Neural Networks can be used in the construction of such graph-comparing functions. Finally, in Sec. 4.3, we present our *inference algorithm* that computes an independent set of any graph $G \in \mathcal{G}$.

### 4.1 MIS Algorithms induced by Graph Comparing Functions

Consider a function $\mathrm{CMP} : \mathcal{G} \times \mathcal{G} \mapsto \{0, 1\}$ that compares two graphs $G, G'$ based on the size of their MIS. Namely, if $|\mathrm{MIS}(G)| \geq |\mathrm{MIS}(G')|$ then $\mathrm{CMP}(G, G') = 0$ and $\mathrm{CMP}(G, G') = 1$ otherwise.

Theorem 1 ensures that if we have access in such a *graph-comparing function* then we can compute an independent set of maximum size for any graph $G$. From a starting node $v$, with an initial graph from $G$, and by recursively selecting either $G/\{v\}$ or $G/\mathcal{N}(v)$ based on $|\mathrm{MIS}(G/\{v\})| \geq |\mathrm{MIS}(G/\mathcal{N}(v))|$, we are ensured to end in an independent set of maximum size. The decision of

---

[1]Note: In the trivial case where $G$ is an empty graph (i.e., it has no edges), the size of the maximum independent set is $|V|$.

---
**Algorithm 1** Randomized Comparator-Induced Algorithm

---
1: **function** $\mathcal{A}^{\mathrm{CMP}}(G(V, E))$                 ▷ Algorithm $\mathcal{A}^{\mathrm{CMP}}(G)$ takes a graph $G$ as input
2:      **if** $|E| = 0$ **then return** $V$
3:      **end if**
4:      pick a vertex $v \in V$ with $d(v) > 0$ uniformly at random.
5:      $G_0 \leftarrow G \setminus \{v\}$ and $G_1 \leftarrow G \setminus \mathcal{N}(v)$
6:      **if** $\mathrm{CMP}(G_0, G_1) = 0$ **then**
7:          $G \leftarrow G_0$                                        ▷ Remove vertex $v$
8:      **else**
9:          $G \leftarrow G_1$                             ▷ Remove the neighbors of $v$
10:     **end if**
11:     **return** $\mathcal{A}^{\mathrm{CMP}}(G)$
12: **end function**

---

whether $|\mathrm{MIS}(G/\{v\})| \geq |\mathrm{MIS}(G/\mathcal{N}(v))|$ at each recursive call can be made according to the output of $\mathrm{CMP}(G/\{v\}, G/\mathcal{N}(v))$.

The cornerstone idea of our approach is that *any graph-comparing function* CMP induces such a recursive algorithm for a MIS. Recursively selecting $G/\{v\}$ or $G/\mathcal{N}(v)$ based on the output of a graph generating function $\mathrm{CMP}(G/\{v\}, G/\mathcal{N}(v)) \in \{0, 1\}$ always guarantees to reach an independent set of the original graph. In case $\mathrm{CMP}(G, G') \neq \mathbb{I}\left[|\mathrm{MIS}(G)| < |\mathrm{MIS}(G')|\right]$, where $\mathbb{I}$ is the indicator function, it is not guaranteed that the computed independent set is of the maximum size. However, there might exist reasonable graph comparing functions that $i$) are efficiently computable $ii$) lead to near-optimal solutions.

In Definition 2 and Algorithm 1 we formalize the idea above.

**Definition 2.** *A comparator* $\mathrm{CMP} : \mathcal{G} \times \mathcal{G} \mapsto \{0, 1\}$ *is a function taking as input two graphs* $G, G'$ *and outputing a* $\{0, 1\}$ *value.*

**Proposition 1.** *Any comparator* $\mathrm{CMP} : \mathcal{G} \times \mathcal{G} \mapsto \{0, 1\}$ *induces a randomized algorithm* $\mathcal{A}^{\mathrm{CMP}}$ *(Algorithm 1).*

**Remark 1.** *Given a graph-comparing function* $\mathrm{CMP} : \mathcal{G} \times \mathcal{G} \mapsto \{0, 1\}$*, the induced algorithm is randomized, since at Step 4 of Algorithm 1, a vertex $v$ is randomly selected. Notice that Algorithm 1 recursively proceeds until a subgraph with $0$ edges is reached (see Step 2).*

**Remark 2.** *Two different comparators* $\mathrm{CMP}$ *and* $\mathrm{CMP}'$ *induce two different algorithms* $\mathcal{A}^{\mathrm{CMP}}$ *and* $\mathcal{A}^{\mathrm{CMP}'}$ *for estimating the maximum independent set.*

### 4.2 Comparators through Graph Neural Networks

In this section, we discuss the architecture of a model $M_\theta : \mathcal{G} \mapsto \mathbb{R}$, parameterized by $\theta \in \Theta$, that is used for the construction of a comparator function

$$\mathrm{CMP}_\theta(G, G') = \mathbb{I}\left[M_\theta(G) < M_\theta(G')\right] .$$

In order to embed graph-level information, we introduce a new GNN module, which we refer to as the Graph Embedding Module (GEM). Unlike standard GNN modules, this module captures different semantic meanings of differing embeddings of a node, its neighbors, and anti-neighbors.

**Graph Embedding Module (GEM):**

The GEM operates using the following recursive formula:

$$\mu_v^{k+1} = \mathrm{LN}\left(\mathrm{GELU}\left(\left[\theta_0^k \mu_v^k \middle\| \theta_1^k \sum_{u \in \mathcal{N}(v)} \mu_u^k \middle\| \theta_2^k \sum_{u \notin \mathcal{N}(v)} \mu_u^k\right]\right)\right) . \tag{1}$$

Initially, all nodes in this graph have zeros embeddings $\mu_v^0 = \vec{0} \in \mathbb{R}^{3p}$. Here, $\mu_v^0$ denotes the initial embedding vector of node $v$. In Eq. (1), for all iterations $k \in [0, \ldots, K-1]$, the embeddings of a

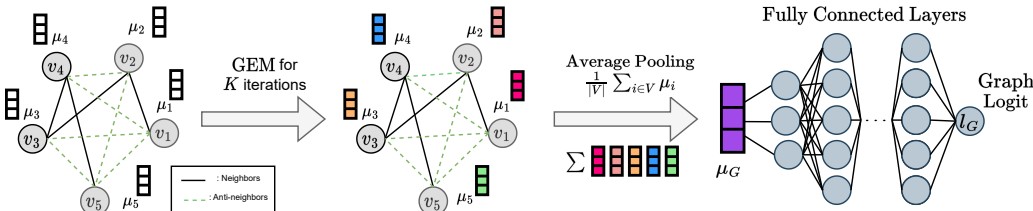

Figure 1: Architecture of model $M_\theta(G)$. From left to right: initially, an input graph $G$ is passed into the model with zeros as node embeddings, which are displayed as white in the figure. The striped green edges connect the anti-neighbors, which are also used in the GEM. After $K$ iterations of the GEM module, the final node embeddings are obtained. These are then averaged to obtain a graph embedding $\mu_G$. Finally, the graph embedding is put through multiple fully-connected layers to obtain a final logit value for the input graph.

node denoted by $\mu_v^k \in \mathbb{R}^{3p}$, its neighbors, and its anti-neighbors $v$ are put through their own linear layers, denoted by $\theta_0^k, \theta_1^k, \theta_2^k \in \mathbb{R}^{p \times 3p}$, which are the parameters of the module. The bias term is omitted in the equation for readability purposes. We incorporate anti-neighbors in the GEM to capture complementary relationships between nodes. By using separate linear layers for different features, we emphasize the contrasting semantic meaning between neighbors and anti-neighbors, representing negative and positive relationships in the graph. Then, the individual feature embeddings are concatenated, which is denoted by $[\ldots \| \ldots]$, followed by a GELU activation function [Hendrycks and Gimpel, 2016] and layer normalization [Ba et al., 2016a]. We note that the computational complexity of this module is $\mathcal{O}(|V|^2)$.

The complete model architecture is depicted in Fig. 1. At a high level, the architecture uses a Graph Embedding Module to extract a global graph embedding from the input graph, which is then passed through a set of fully connected layers to output a logit for that graph. During the training process of the comparator function, we utilize $\text{CMP}_\theta(G, G') = \text{softmax}\left([M_\theta(G) \| M_\theta(G')]\right)$, which forms a differentiable loss function for classification.

### 4.3 Inference Algorithm

In the previous section, we discussed how a parameterization $\theta \in \Theta$ defines the graph-comparing function $\text{CMP}_\theta(G, G') = \mathbb{I}\left[M_\theta(G) < M_\theta(G')\right]$. As a result, the same parameterization $\theta \in \Theta$ defines an algorithm $\mathcal{A}^{\text{CMP}_\theta}$, where at Step 6 of Algorithm 1, the comparing function $\text{CMP}_\theta$ is used. Since the number of graph comparisons is upper-bounded by $|V|$, the inference algorithm admits to a computational complexity of $\mathcal{O}(|V|^3)$.

## 5 Self-Supervised Training through the Consistency Property

In this section, we present our methodology for selecting the parameters $\theta \in \Theta$ so that the resulting inference algorithm $\mathcal{A}^{\text{CMP}_\theta}(\cdot)$ computes independent sets with (close to) the maximum value.

The most straightforward approach is to select the parameters $\theta \in \Theta$ such that $\text{CMP}_\theta(G, G') \simeq \mathbb{I}\left[|\text{MIS}(G)| < |\text{MIS}(G')|\right]$ using labeled data. The problem with this approach is that a huge amount of annotated data of the form $\{((G, G'), \mathbb{I}\left[|\text{MIS}(G)| < |\text{MIS}(G')|\right])\}$ are required. Since finding the MIS is an NP-Hard problem, annotating such data comes with an insurmountable computational burden.

The **key idea** to overcome the latter limitation is to annotate the data of the form $\{(G, G')\}$ by using the algorithm $\mathcal{A}^{\text{CMP}_\theta}(\cdot)$ that runs in polynomial time with respect to the size of the graph. Intuitively, our proposed framework entails the optimization of the parameterized comparator function $\text{CMP}_\theta$ on data generated using algorithm $\mathcal{A}^{\text{CMP}_\theta}$. A better comparator function leads to a better algorithm, which leads to better data, and vice versa. This mutually reinforcing relationship between the two components of our framework is theoretically indicated by Theorem 2 that we present in Section 5.1. The exact steps are detailed below.

## 5.1 Consistent Graph Comparing Functions

In this section, we introduce the notion of a *consistent* graph-comparing function (Definition 3) that plays a critical role in our self-supervised learning approach. Kindly take note that $\mathcal{A}^{\mathrm{CMP}}$ utilizes the unparameterized variant of a comparator function, whereas $\mathcal{A}^{\mathrm{CMP}_\theta}$ utilizes its parameterized counterpart.

**Definition 3** (Consistency). *A graph-comparing function* $\mathrm{CMP} : \mathcal{G} \times \mathcal{G} \mapsto \{0, 1\}$ *is called consistent if and only if for any pair of graphs* $G, G' \in \mathcal{G}$,

$$\mathrm{CMP}(G, G') = 0 \text{ if and only if } \mathbb{E}\left[\left|\mathcal{A}^{\mathrm{CMP}}(G)\right|\right] \geq \mathbb{E}\left[\left|\mathcal{A}^{\mathrm{CMP}}(G')\right|\right].$$

**Remark 3.** *In Definition 3 we use* $\mathbb{E}\left[\left|\mathcal{A}^{\mathrm{CMP}}(G)\right|\right], \mathbb{E}\left[\left|\mathcal{A}^{\mathrm{CMP}}(G')\right|\right]$ *since, as we have already discussed, a comparator* $\mathrm{CMP}$ *induces a randomized algorithm* $\mathcal{A}^{\mathrm{CMP}}$*, where the expectation is over the nodes in the graphs.*

In Theorem 2, we formally establish that any *consistent graph-comparing function* CMP induces an optimal algorithm for the MIS.

**Theorem 2.** *Consider a consistent comparator* $\mathrm{CMP} : \mathcal{G} \times \mathcal{G} \mapsto \{0, 1\}$. *Then, the algorithm* $\mathcal{A}^{\mathrm{CMP}}(\cdot)$ *always computes a Maximum Independent Set,* $\mathbb{E}\left[\left|\mathcal{A}^{\mathrm{CMP}}(G)\right|\right] = |\mathrm{MIS}(G)|$ *for all* $G \in \mathcal{G}$.

Theorem 2 guarantees that any consistent graph comparing function CMP induces an optimal algorithm $\mathcal{A}^{\mathrm{CMP}}$ for MIS. The proof for this theorem can be found in Appendix I. Hence, the selection of parameters $\theta^\star \in \Theta$ should be selected such that $\mathrm{CMP}_{\theta^\star}$ is *consistent*. More precisely:

**Goal of Training:** Find parameters $\theta^\star \in \Theta$ such that for all $G, G' \in \mathcal{G}$:

$$\mathrm{CMP}_{\theta^\star}(G, G') = 0 \text{ if and only if } \mathbb{E}\left[\left|\mathcal{A}^{\mathrm{CMP}_{\theta^\star}}(G)\right|\right] \geq \mathbb{E}\left[\left|\mathcal{A}^{\mathrm{CMP}_{\theta^\star}}(G')\right|\right].$$

## 5.2 Training a Consistent Comparator

The cornerstone idea of our self-supervised learning approach is to make the comparator more and more consistent over time. Namely, the idea is to update the parameters as follows:

$$\theta^{t+1} := \mathrm{argmin}_{\theta \in \Theta} \mathbb{E}_{G, G'}\left[\ell\left(\mathrm{CMP}_\theta(G, G'), \mathbb{I}\left[\mathbb{E}\left[\left|\mathcal{A}^{\mathrm{CMP}_{\theta_t}}(G)\right|\right] < \mathbb{E}\left[\left|\mathcal{A}^{\mathrm{CMP}_{\theta_t}}(G')\right|\right]\right]\right)\right], \tag{2}$$

where $\ell(\cdot, \cdot)$ is a binary classification loss. In Eq. (2), $\theta_t$ are the fixed parameters of the previous epoch. Thus, in the next few paragraphs, we only use the notation $\mathcal{A}^{\mathrm{CMP}_{\theta_t}}$ to denote the fixed parameters. Gradient updates are only computed over $\theta$.

**Remark 4.** *We remark that neither solving the non-convex minimization problem of Eq.* (2) *nor the existence of parameters* $\theta^\star \in \Theta$ *such that* $\mathrm{CMP}_{\theta^\star}$ *can be guaranteed. However, using a first-order method for Eq.* (2) *and a large enough parameterization can lead to an approximately consistent comparator with approximately optimal performance.*

In Algorithm 2, we present the basic pipeline of the self-training approach that selects the parameters $\theta \in \Theta$ such that the inference algorithm $\mathcal{A}^{\mathrm{CMP}_{\theta_t}}$ admits a competitive performance given as input graphs $G$ following a graph distribution $\mathcal{D} \subseteq \mathcal{G}$. We further improve this basic pipeline with several tweaks that we incorporate into our training process:

**Creating a graph buffer $\mathcal{B}$:** We are given a shuffled dataset of graphs $\mathcal{D}$, which represents the training data for the model. The core difference between the pipeline and the training process comes from the graph buffer $\mathcal{B}$. In Algorithm 2, this buffer stores any graph $G$ that is found during the recursive call of $\mathcal{A}^{\mathrm{CMP}_{\theta_t}}(G_{\mathrm{init}})$ on $G_{\mathrm{init}} \sim D$ (Step 6 of Algorithm 2). However, in the implementation of the graph buffer, it stores pairs of graphs $(G, G')$ that were generated by $\mathcal{A}^{\mathrm{CMP}_{\theta_t}}(G_{\mathrm{init}})$, alongside a binary label that indicates which of the two graphs has a larger estimated MIS size. How this estimate is generated, will be explained further down this section.

**The training process:** Prior to starting training, we first set two hyperparameters: one that specifies the number of graphs used to populate the buffer before training the model, and another that determines the number of graph pairs generated from $\mathcal{A}^{\mathrm{CMP}_{\theta_t}}(G)$ per graph $G \sim \mathcal{D}$. Then, a dataset is created by generating these pairs for the set number of graphs. The dataset is then added to the graph buffer,

---

**Algorithm 2** Basic Pipeline of our Training Approach

---

1: **Input:** A distribution $\mathcal{D}$ over graphs.
2: Initialize parameters $\theta_0 \in \Theta$.
3: Initialize a *graph-buffer* $\mathcal{B} \leftarrow \varnothing$.
4: **for** each epoch $t = 0, \ldots, T - 1$ **do**
5:     Sample a graph $G_{\text{init}} \sim \mathcal{D}$.
6:     Run $\mathcal{A}^{\text{CMP}_{\theta_t}}(G_{\text{init}})$ and store in $\mathcal{B}$ all graphs produced during each recursive call of Algorithm 1.
7:     Update the parameters $\theta_{t+1} \in \Theta$ such that

$$\theta^{t+1} := \text{argmin}_{\theta \in \Theta} \mathbb{E}_{(G,G') \sim \mathcal{B}} \left[ \ell \left( \text{CMP}_\theta(G, G'), \mathbb{I}\left[ \mathbb{E}\left[ |\mathcal{A}^{\text{CMP}_{\theta_t}}(G)| \right] < \mathbb{E}\left[ |\mathcal{A}^{\text{CMP}_{\theta_t}}(G')| \right] \right] \right) \right]$$

8: **end for**

---

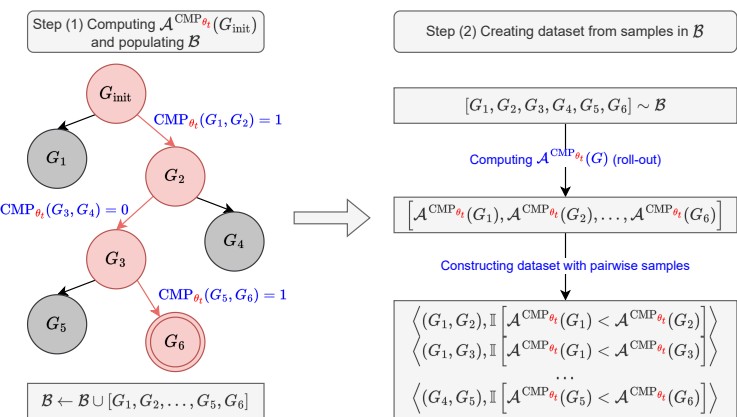

Figure 2: An example of data generation for the training. (Left) At the beginning of each training epoch (Step 5), Algorithm 2 samples $G_{\text{init}} \sim \mathcal{D}$ and computes an independent set by using the comparator $\text{CMP}_{\theta_t}$ and by following the branches in the *recursion tree* that are marked red, with the doubly circled one being the produced independent set. The generated graphs from this procedure are added to the buffer $\mathcal{B}$. (Right) Then, a dataset is created by sampling graphs from the buffer and then computing an estimate of their MIS size (based on $\mathcal{A}^{\text{CMP}_{\theta_t}}$). Based on this estimate, a dataset is created with graph pairs $(G, G')$ and their corresponding binary labels denoting which MIS estimates are larger.

replacing steps 5 and 6 in Algorithm 2, which only does this with one graph per epoch. Next, training starts, and after completing a set number of epochs, a new dataset is created using the updated model, and the process is repeated iteratively.

**Estimating the MIS:** Finally, the loss function in Step 7 of Algorithm 2, also operates slightly differently. The main difference arises from $\mathbb{I}\left[ \mathbb{E}\left[ |\mathcal{A}^{\text{CMP}_{\theta_t}}(G)| \right] < \mathbb{E}\left[ |\mathcal{A}^{\text{CMP}_{\theta_t}}(G')| \right] \right]$, since the estimates $|\text{MIS}(G)| \approx \mathbb{E}\left[ |\mathcal{A}^{\text{CMP}_{\theta_t}}(G)| \right]$ and $|\text{MIS}(G')| \approx \mathbb{E}\left[ |\mathcal{A}^{\text{CMP}_{\theta_t}}(G')| \right]$ are not directly utilized. Instead, we propose two other approaches, which are better approximations than the expectations used in Algorithm 2, since they use a maximizing operator.

The first approach involves performing so-called "roll-outs" on the graph pairs generated $G$ and $G'$ by $\mathcal{A}^{\text{CMP}_{\theta_t}}$, in order to estimate their MIS sizes. To perform the roll-outs, we simply run $\mathcal{A}^{\text{CMP}_{\theta_t}}$ on graphs $G$ and $G'$ $m$ times and use the maximum size of the found independent sets as an estimate of their MIS. Formally, in a roll-out on a graph $G$, we sample the independent sets $\text{ISS}_1, \text{ISS}_2, \ldots, \text{ISS}_m \sim \mathcal{A}^{\text{CMP}_{\theta_t}}(G)$. Then, the estimate of the MIS size of $G$ is $\max(|\text{ISS}_1|, |\text{ISS}_2|, \ldots, |\text{ISS}_m|)$.

An example of the entire process of generating the dataset using roll-outs can be found in Fig. 2. In practice, we observe an increasing consistency of the model during training, as can be seen in Appendix J.

**Mixed roll-out variant**: We introduce a variant of the aforementioned method, which utilizes the deterministic greedy algorithm. This greedy algorithm iteratively creates an independent set by removing the node with the lowest degree and adding it to the independent set. This algorithm is often an efficient approximation to the optimum solution. Our variant is constructed as follows: we compute the maximum between the roll-outs of the model and the result of the greedy algorithm, which creates a dataset with more accurate self-supervised approximations of the MIS values. This, in turn, generates binary targets for the buffer that are more likely to be accurate. Thus, for this second variant, the estimate of the MIS size of a graph $G$ would be $\max\left(|\text{Greedy}(G)|, |\text{ISS}_1|, |\text{ISS}_2|, \ldots, |\text{ISS}_m|\right)$.

## 6 Experiments

In this section, we conduct an evaluation of the proposed method for the MIS problem. Let us first describe the training setup, the baselines, and the datasets. Additional details and experiments on MIS are displayed in the Appendices C and D. Our method also generalizes well in MVC, as the results in Appendix E illustrate.

### 6.1 Training Setup

**Our model**: We implement two comparator models: one using just roll-outs with the model, and another using the roll-outs together with greedy, called "mixed roll-out". We train each model using a graph embedding module with $K = 3$ iterations, which takes in 32-dimensional initial node embeddings.

**Baselines**: We compare against the neural approaches *Erdos GNN* [Karalias and Loukas, 2020], *RUN-CSP* from Toenshoff et al. [2019], and a method specifically for the MIS problem: LwDMIS [Ahn et al., 2020]. Since we observe unexpected poor performances from RUN-CSP on the COLLAB and RB datasets, we have omitted those results from the table. We train every model for 300 epochs. Each experiment is performed on a single GPU with 6GB RAM.

Besides neural approaches, we use traditional baselines, such as the *Greedy MIS* [Wormald, 1995] , *Simple Local Search* [Feo et al., 1994] and a *Random Comparator* as a sanity check. Furthermore, we implement two mixed-integer linear programming solvers: *SCIP 8.0.3* and the highly optimized commercial solver *Gurobi 10.0*.

**Datasets**: We evaluate our model on three standard datasets, following Karalias and Loukas [2020]: COLLAB [Yanardag and Vishwanathan, 2015], TWITTER [Leskovec and Krevl, 2014] and RB [Xu et al., 2007, Toenshoff et al., 2019]. In addition, we introduce the SPECIAL dataset that includes challenging graphs for handcrafted approaches as we detail in Appendix C.

### 6.2 Results

Table 1 reports the average approximation ratios on the test instances of the various datasets. The approximation ratio is computed by dividing a solution's independent set size by the optimum solution, which is computed using the Gurobi solver with a time limit of 1 hour per graph.

The results indicate that the greedy algorithm performs strongly in three of the four datasets, which is consistent with the observation of Angelini and Ricci-Tersenghi [2022]. However, notice that our proposed approach outperforms the greedy in both the Twitter and the SPECIAL datasets, which validates that the greedy heuristic is not optimal in every case and is prone to failing in few cases. Importantly, among the neural approaches that are the main compared methods, our proposed method performs favorably in all datasets. The performance of our method indicates that the proposed self-training scheme is able to learn from diverse data distributions and generalize reasonably well in the test sets of the respective dataset. In addition, the proposed method is faster than the rest neural approaches.

The mixed roll-out model in Table 1 outperforms the normal roll-out model in almost all datasets, indicating the effectiveness of the greedy heuristic in roll-outs. This is particularly evident in the RB dataset. However, for SPECIAL instances, the normal model performs marginally better, possibly due to the unsuitability of the greedy guiding heuristic as a baseline for this dataset.

**Out of distribution:** We examine the performance of the learned comparator through its generalization to new graph distributions. Concretely, we conduct an out-of-distribution analysis as follows:

Table 1: Test set approximation ratios (higher is better; best performance in bold) on four datasets for MIS. We report the average approximation ratios, along with std and duration in seconds per graph (s/g). Notice that the proposed method outperforms all the deep-learning-based approaches across datasets. Classical methods have a gray color, whilst neural approaches are in black.

| Method (↓) Dataset (→) | RB | COLLAB | TWITTER | SPECIAL |
|---|---|---|---|---|
| CMP (Normal Roll-outs) | $0.770 \pm 0.107$ (0.43 s/g) | $0.990 \pm 0.051$ (0.17 s/g) | $0.967 \pm 0.083$ (0.35 s/g) | $\mathbf{0.996 \pm 0.029}$ (0.04 s/g) |
| CMP (Mixed Roll-outs) | $\mathbf{0.836 \pm 0.083}$ (0.36 s/g) | $\mathbf{0.990 \pm 0.049}$ (0.21 s/g) | $\mathbf{0.977 \pm 0.031}$ (0.21 s/g) | $0.994 \pm 0.035$ (0.05 s/g) |
| Erdos' GNN | $0.813 \pm 0.107$ (1.39 s/g) | $0.952 \pm 0.142$ (0.60 s/g) | $0.935 \pm 0.078$ (1.37 s/g) | $0.921 \pm 0.218$ (1.03 s/g) |
| LwDMIS | $0.804 \pm 0.089$ (0.42 s/g) | $0.978 \pm 0.031$ (0.17 s/g) | $0.972 \pm 0.032$ (0.19 s/g) | $0.828 \pm 0.304$ (0.32 s/g) |
| RUN-CSP (Accurate) | − | − | $0.875 \pm 0.053$ (0.57 s/g) | $0.946 \pm 0.059$ (0.51 s/g) |
| Greedy MIS | $0.925 \pm 0.053$ (0.01 s/g) | $0.998 \pm 0.023$ (0.02 s/g) | $0.964 \pm 0.048$ (0.04 s/g) | $0.131 \pm 0.055$ (0.03 s/g) |
| Random CMP | $0.615 \pm 0.155$ (0.42 s/g) | $0.817 \pm 0.211$ (0.30 s/g) | $0.634 \pm 0.182$ (0.36 s/g) | $0.225 \pm 0.279$ (0.41 s/g) |
| Simple Local Search (10s) | $0.565 \pm 0.237$ | $0.860 \pm 0.213$ | $0.644 \pm 0.218$ | $0.188 \pm 0.340$ |
| SCIP 8.0.3 (1s) | $0.741 \pm 0.351$ | $0.999 \pm 0.016$ | $0.959 \pm 0.024$ | 1.000 |
| SCIP 8.0.3 (5s) | $0.937 \pm 0.118$ | 1.000 | $0.999 \pm 0.024$ | 1.000 |
| Gurobi 10.0 (0.5s) | $0.969 \pm 0.070$ | $0.981 \pm 0.068$ | $0.985 \pm 0.085$ | $0.940 \pm 0.237$ |
| Gurobi 10.0 (1s) | $0.983 \pm 0.051$ | 1.000 | 1.000 | 1.000 |
| Gurobi 10.0 (5s) | $0.999 \pm 0.008$ | 1.000 | 1.000 | 1.000 |

Table 2: Out-of-distribution approximation ratios during inference (higher is better). Every row denotes a model trained on a specific dataset. Every column considers a different test dataset. The CMP is trained using mixed roll-outs. Notice that the proposed method generalizes well in out-of-distribution structures. This is indicative of the learned comparator extracting robust patterns.

| Model (↓) Dataset (→) | RB | COLLAB | TWITTER |
|---|---|---|---|
| CMP RB | − | $0.903 \pm 0.186$ | $0.668 \pm 0.187$ |
| CMP COLLAB | $0.856 \pm 0.080$ | − | $0.906 \pm 0.094$ |
| CMP TWITTER | $0.773 \pm 0.101$ | $0.927 \pm 0.148$ | − |
| Erdos' GNN RB | − | $0.361 \pm 0.334$ | $0.752 \pm 0.188$ |
| Erdos' GNN COLLAB | $0.680 \pm 0.071$ | − | $0.592 \pm 0.186$ |
| Erdos' GNN TWITTER | $0.746 \pm 0.092$ | $0.666 \pm 0.385$ | − |

each model is trained in one graph distribution, indicated by the rows of Table 2. Then, the model is evaluated on different graph distributions, indicated by the columns of Table 2. The analysis is conducted on both our model and the approach of *Erdos GNN* [Karalias and Loukas, 2020].

Surprisingly, our model trained over COLLAB displays good generalization skills across different datasets, even outperforming the RB-trained model on the RB dataset. Conversely, *Erdos GNN* trained over RB performs poorly over the COLLAB dataset. Both models trained over the RB dataset perform more poorly in general, likely due to the highly specific graph distribution of the RB dataset. Moreover, our model, on the whole, exhibits good generalization skills over different graph distributions.

# 7 Conclusion

Motivated by the principles of Dynamic Programming, we develop a self-training approach for important CO problems, such as the Maximum Independent Set and the Minimum Vertex Cover. Our approach embraces the power of self-training, offering the dual benefits of data self-annotation and data generation. These inherent attributes are instrumental in providing an unlimited source of data indicating that the performance of the induced algorithms can be significantly improved with sufficient scaling on the computational resources. We firmly believe that a thorough investigation into the interplay between Dynamic Programming and self-training techniques can pave the way for new deep-learning-oriented approaches for demanding CO problems.

**Limitations**: Our current empirical approach lacks theoretical guarantees on the convergence or the approximate optimality of the obtained algorithm. Additionally, the implemented GNN is using simple modules, while more complex modules could result in further empirical improvements, which can be the next step in this direction. Furthermore, randomly selecting a vertex could be sub-optimal. One interesting future direction would be to explore predicting the next vertex to select.

## Acknowledgements

This work was supported by the Swiss National Science Foundation (SNSF) under grant number 200021_205011, by Hasler Foundation Program: Hasler Responsible AI (project number 21043) and Innovation project supported by Innosuisse (contract agreement 100.960 IP-ICT). We further thank Elias Abad Rocamora and Nikolaos Karalias for the useful feedback on the paper draft as well as Planny for the helpful discussions.

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

# Contents of the Appendix

We describe the contents of the supplementary material below:

- In Appendix A, we provide further information on the prior work and related research, including additional references and links to relevant papers.
- In Appendix B, we explore applications of our method beyond the MIS problem. Specifically, we discuss how our method can be applied to the Minimum Vertex Cover (MVC) problem.
- In Appendix C, we provide detailed experimental details used in our experiments.
- In Appendix D, we present additional experiments conducted specifically on the MIS problem.
- In Appendix E, we extend our investigations to the MVC problem. We present experimental results on benchmarks and discuss the performance and insights gained from the experiments.
- Appendix F provides additional background information on relevant concepts related to the MIS and MVC problems.
- In Appendix G, we present an ablation study where we systematically analyze the impact of different architecture configurations in our method.
- Appendix H delves into the broader impact of our work, discussing the potential implications of using our self-supervised approach for CO problems.
- In Appendix I, we provide the omitted proofs of theorems mentioned in the main paper.
- Finally, in Appendix J, we gain some insight into the decisions of the comparator through the consistency property.

## A  Additional Links to Prior Work

In Appendix A.1, we delve into an analysis of multiple deep learning techniques for solving CO (Combinatorial Optimization) problems, including how different works have addressed the non-continuity of the loss function. Additionally, in Appendix A.2 we provide a brief explanation of Dynamic Programming and Monte Carlo Tree Search.

### A.1  Deep Learning Approaches for CO

The growing interest in employing machine learning techniques to tackle combinatorial optimization problems has led to addressing the task using different neural approaches [Karimi-Mamaghan et al., 2022, Mazyavkina et al., 2021]. Moreover, some works combine elements from classic heuristics with deep learning techniques, like *branch-and-bound* [Gasse et al., 2019] and *local search* [Li et al., 2018].

As we mention in Sec. 2, several works use a continuous relaxation of the loss function. Following the settings of Wang and Li [2023], CO problems consist of assigning values to discrete optimization variables $X = (x_i)_{1 \leq i \leq n} \in \{0, 1\}^n$ such that an energy (or loss) function $F(G, X)$ is minimized or maximized in $X$. Since $F(G, X)$ is not continuous, Karalias and Loukas [2020] employs a relaxed probabilistic loss function, which is later refined by Wang et al. [2022] by assuming continuous values and taking the values of $F(G, X)$ at the discrete points $X$. Since some CO problems require *instance-wise good solutions*, Wang and Li [2023] focuses more on obtaining good initialization instances. Given that certain CO issues necessitate the acquisition of *individual-level optimal solutions*, Wang and Li [2023] places greater emphasis on attaining favorable initializations for subsequent instances rather than providing immediate solutions.

### A.2  Dynamic Programming

Dynamic Programming is especially effective for solving problems that demonstrate the principle of optimality. This principle states that an optimal solution to a problem can be achieved by recursively finding the optimal solutions to smaller subproblems.

Xu et al. [2020] utilize DNNs to replace the function responsible for dividing a problem into sub-problems and estimating the optimal decision at each step. Their results demonstrate significant

reductions in computation time for classical combinatorial optimization problems such as the 'TSP', compared to the Bellman-Held-Karp algorithm [Bellman, 1962].

The work of Yang et al. [2018] addresses the problem of the size of the lookup table, which may require exponential space, used to store solutions of sub-problems in Dynamic Programming. In their work, Yang et al. [2018] argue that it is possible to approximate a Dynamic Programming function with a much smaller neural network.

The Dynamic Programming Tree (DPT) is often employed for solving CO tasks. Each node of the tree represents a problem, and its child nodes represent two sub-problems derived from it. The DPT is designed to explore the solution space of a problem similar to Dynamic Programming, but it only considers a subset of the possible solutions due to the large size of the solution space. To construct the DPT, a Monte Carlo Tree Search (MCTS) technique [James et al., 2017] is utilized. The MCTS algorithm begins at the root node and navigates the tree to add new nodes to the structure. Upon reaching a new leaf node, MCTS employs a simulation to estimate the value of the node. The results of these simulations are subsequently backpropagated up through the tree, updating the node values accordingly.

Due to the reasons stated above, Monte Carlo Tree Search (MCTS) and other tree search methods have been utilized in the context of CO problems as a replacement for pure Dynamic Programming. MCTS is capable of significantly reducing computational complexity as it operates on a smaller search space. In Xing et al. [2020], MCTS is employed to solve the TSP problem on graphs with a maximum of 100 nodes. Additionally, Xu and Lieberherr [2020] employs MCTS to convert CO problems into Zermelo games [Schwalbe and Walker, 2001], and maps the winning strategy of the Zermelo games to provide solutions for CO problems.

## B   Beyond Maximum Independent Set

Our proposed method for solving CO problems using DP and GNNs with self-supervised training extends beyond the realm of the MIS problem. While our focus has primarily been on MIS, the underlying framework and techniques can be applied to various other CO settings. In this section, we explore the potential application and adaptation of our approach to address a different problem domain. We will focus on another problem, called the Minimum Vertex Cover (MVC) problem. We will benchmark against various neural approaches. The goal of the MVC problem is the following:

**Definition 4.** *Given a graph $G(V, E)$ find a set of nodes $S \subseteq V$ with minimum size such that for each edge $(v, u) \in E$ either $v \in S$ or $u \in S$.*

Minimum Vertex Cover asks for the smallest set of vertices $S$ such that all edges of $G$ are "covered" by at least one vertex in the set. Dynamic Programming provides similar optimality guarantees as that of Maximum Independent Set described in Theorem 1. The respective optimality guarantees for Minimum Vertex Cover are formally stated in Corollary 1.

**Corollary 1.** *Let a graph $G(V, E) \in \mathcal{G}$. Then for any vertex $v \in V$ with $d(v) \geq 1$,*

$$|\mathrm{MVC}(G)| = \min\left(|\mathrm{MVC}\left(G_0\right)|, |\mathrm{MVC}(G_1)|\right)^{[2]}.$$

*where*

- $G' := G(V \setminus \{v\}, E \setminus \{(u, \ell) \mid u \in \mathcal{N}(v) \wedge \ell \in \mathcal{N}(u)\})$, $G'$ *is created by removing $v$ from $G$ as well as all the edges incident to neighbors of $v$.*

- $G_0 := G'(V \cup \{u'|u \in \mathcal{N}(v)\}, E \cup \{(u', u)|u \in \mathcal{N}(v)\})$, $G_0$ *is created from $G'$ by adding the copy nodes $u'|\forall u \in \mathcal{N}(v)$ and the edges $(u', u)$ for all $u \in \mathcal{N}(v)$ (each vertex $u$ is connected with its copy $u'$). $G_0$ represents the situation where vertex $v$ is not selected in the MVC, but its neighbors are.*

- $G_1 = G(V \cup \{v'\}, E \setminus \{(u, v)|u \in \mathcal{N}(v)\} \cup \{(v', v)\})$, $G_1$ *is created from $G$ by removing all edges incident to $v$ and adding a copy node $v'$ that is then connected to $v$. $G_1$ represents the situation where $v$ is selected to be part of the MVC.*

---

[2]In the trivial case where $G$ is a graph with only connected components of two vertices ($d(v) \leq 1$ for all $v \in V$), the size of the minimum vertex cover is $|G| = |E(G)|$.

Corollary 1 is based on the fact that for any node $v \in V$ either $v$ or *all of its neighbors* $\mathcal{N}(v)$ lie in the optimal solution. The first case is captured through $G_1$. Notice that $G_1$ is constructed by $G$ by removing all the incident edges to $v \in V$ since they are already covered by $v$. The new copy node $v'$ is added and connected to $v$ so as to encode that either node $v$ or its copy $v'$ should be selected in the minimum vertex cover of $G_1$. Symmetrically, the graph $G_0$ is constructed by repeating the latter process for each $u \in \mathcal{N}(v)$.

As in Algorithm 1, a graph comparing function $\mathrm{CMP}_{\mathrm{MVC}}(G, G')$ induces the following recursive algorithm for computing a vertex cover. The latter is formally described in Algorithm 3. By Corollary 1 in case $\mathrm{CMP}_{\mathrm{MVC}}(G, G') = \mathbb{I}\left[|\mathrm{MVC}(G)| > |\mathrm{MVC}(G')|\right]$ [3] then Algorithm 3 always computes a minimum vertex cover.

---

**Algorithm 3** Comparator-Induced Algorithm for the MVC problem

---

1: **function** $\mathcal{A}^{\mathrm{CMP}_{\mathrm{MVC}}}(G(V, E))$      $\triangleright$ Algorithm $\mathcal{A}^{\mathrm{CMP}_{\mathrm{MVC}}}(G)$ takes a graph $G$ as input
2:   **if** $\forall v \in V : d(v) \leq 1$ **then**     $\triangleright$ $G$ is composed of isolated nodes and isolated edges
3:    $S \leftarrow \varnothing$
4:    **for** each edge $(v, v') \in E$ **do**
5:     $S \leftarrow S \cup \{v\}$       $\triangleright$ Select one of the endpoints of each isolated edge
6:    **end for**
7:    **return** $S$
8:   **end if**
9:   pick a vertex $v \in V$ with $d(v) \geq 1$ uniformly at random.
10:   $G' := G(V \setminus \{v\}, E \setminus \{(u, \ell) \mid u \in \mathcal{N}(v) \land \ell \in \mathcal{N}(u)\})$
11:   $G_0 := G'(V \cup \{u' | u \in \mathcal{N}(v)\}, E \cup \{(u', u) | u \in \mathcal{N}(v)\}$
12:   $G_1 = G(V \cup \{v'\}, E \setminus \{(u, v) | u \in \mathcal{N}(v)\} \cup \{(v', v)\})$
13:   **if** $\mathrm{CMP}_{\mathrm{MVC}}(G_0, G_1) = 0$ **then**
14:    $G \leftarrow G_0$        $\triangleright$ Remove vertex $v$ and put it's neighbors in the MVC
15:   **else**
16:    $G \leftarrow G_1$     $\triangleright$ Put vertex $v$ in the MVC and remove the edges to its neighbors
17:   **end if**
18:   **return** $\mathcal{A}^{\mathrm{CMP}_{\mathrm{MVC}}}(G)$
19: **end function**

---

By adjusting the notion of consistency (see Definition 3) in the context of minimum vertex cover as $\mathrm{CMP}_{\mathrm{MVC}}(G, G') = 0$ if and only if $\mathbb{E}\left[\left|\mathcal{A}^{\mathrm{CMP}_{\mathrm{MVC}}}(G)\right|\right] \leq \mathbb{E}\left[\left|\mathcal{A}^{\mathrm{CMP}_{\mathrm{MVC}}}(G')\right|\right]$, one can establish that, if $\mathrm{CMP}_{\mathrm{MVC}}$ is a consistent comparator, $\mathcal{A}^{\mathrm{CMP}_{\mathrm{MVC}}}(\cdot)$ always outputs a minimum vertex cover. In Appendix E, we present our experimental results produced after the self-training of a consistent in the context of Minimum Vertex Cover.

# C   Experimental Details

Additional information related to the hyper-parameters, baselines, and datasets are depicted in this section.

## C.1   Training Setup and Baselines

**Training setup**: The sizes of the linear layers at each iteration are as follows: $96 \times 32$, $96 \times 32$, and $96 \times 32$. After this module, $4$ linear layers follow, with the following sizes: $96 \times 32$, $32 \times 32$, $32 \times 32$, and $64 \times 1$. Furthermore, the output of the last GEM iteration has a skip connection into the last dense layer of the model, which is also why the final layer has $64$ input neurons. Every linear layer in the entire network is followed by layer normalization [Ba et al., 2016b]. Each model is trained for 300 epochs total with batch size 32 using Adam optimizer [Kingma and Ba, 2014] and a learning rate of $0.001$. We split the graph datasets into $80\%$ for training and $20\%$ for testing, except

---

[3]Notice that for the MVC, we have a $>$ sign in the function instead of a $<$ sign like in the MIS. This comes from the fact that the recursive algorithm for the MVC in corollary 1 aims to find a minimum-sized set, instead of the maximum like in Theorem 1.

for COLLAB and RB datasets where the number of test graphs is much bigger than the number of train graphs. From the training data, a validation set is constructed by dedicating $20\%$ of the data in the buffer to it. Considering the baseline methods, we trained them for 300 epochs, and in the case of reinforcement learning, for 300 episodes. A precise description of the Hyper-parameters values is reported in Table 3.

**Baselines**: Besides the methods specified in Sec. 6, in Appendix D we employ the IMDB dataset [Morris et al., 2020] and graphs coming from the *Erdos-Renyi* [Erdős et al., 1960], *Barabasi-Albert* [Albert and Barabási, 2002] and *Watts-Strogatz* [Watts and Strogatz, 1998] distributions. Moreover, in Appendix E we use two datasets (RB200 and RB500) that resemble the RB dataset employed in Sec. 6, but the two datasets have different numbers of nodes. As mentioned in Sec. 6 and in Appendix E, we use traditional baselines, such as the *Greedy MIS* employed in Sec. 6 and *Greedy MVC* employed in Appendix E. *Greedy MIS* iteratively chooses the node with the minimum degree to be part of the independent set and removes its neighbors from the graph, while *Greedy MVC* considers the node with the highest degree as part of the vertex cover set, still removing the neighbors of the node. We also implement *Simple Local Search*, which, given a time limit, randomly adds a node to the independent set and removes conflicting nodes. The largest set in the time period is used as the solution of the algorithm.

**Hardware**: Any experiment with a reported execution time has been performed on a laptop with Windows 10 and Python 3.10. The device uses a CUDA-enabled NVIDIA GeForce GTX 1060 Max-Q GPU with 6GB VRAM, and an Intel(R) Core(TM) I7-7700HQ CPU with a 2.80GHz clock speed, and 16GB of RAM. Our method is implemented in PyTorch.

## C.2 Datasets

Two important parameters are considered for the analysis of the datasets: the size and the density of the graphs. The first parameter is measured in terms of total number of nodes. The second one reflects how sparse a graph is and it is measured as the ratio between the total number of edges of the graph and the total number of edges if the graph would have been complete.

**Real-world datasets and RB**: Our work utilizes real-world datasets that simulate social network scenarios, like IMDB, COLLAB and TWITTER. The RB dataset that we employ consists of the same graphs utilized in the research conducted by Karalias and Loukas [2020], whereas the RB200 and RB500 datasets align with those employed in the work by Wang and Li [2023]. Notably, RB200 and RB500 were generated by configuring a small hyper-parameter $\rho = 0.25$, making the instances within these datasets challenging, as specified by Wang and Li [2023].

**Graph distribution**: In Appendix D, we conduct testing on well-known graph distributions, specifically the *Erdos-Renyi*, *Barabasi-Albert*, and *Watts-Strogatz* distributions [Erdős et al., 1960, Albert and Barabási, 2002, Watts and Strogatz, 1998], abbreviated as ER, BA, and WS in Table 4. The objective is to evaluate the performance of our model on datasets where the instance densities differ from those of real-world datasets.

**Special dataset**: The *Special* dataset has two nodes, $u$ and $v$, which are connected to a set of nodes $I$. The size of set $I$ is denoted as $n$. Moreover, the nodes within $I$ are all independent of each other, since there are no connections between any pair of nodes $i$ and $j$ within $I$. Additionally, all nodes in $I$ are connected to every node in a clique of nodes called $C$, which has a total of $n + a$ nodes. When employing the Greedy MIS heuristic, nodes $u$, $v$, and one node from $C$ are selected as part of the MIS (Maximum Independent Set), while the MIS itself in reality corresponds to $I$, which has a dimensionality of $n$. Notice that the Greedy MIS heuristic performs poorly in these instances.

Table 3: Experimental setting employed in the training process.

| Name | Value |
|---|---|
| Learning rate | 0.001 |
| Optimizer | Adam |
| Output layer size (D) | 32 |
| Number of GEM layers (K) | 3 |
| Number of fully connected layers (L) | 4 |
| Batch size | 32 |
| Training epochs | 300 |

Table 4: Statistics of the employed datasets. For each dataset, it is reported the average number of nodes and density of the graphs, and the total number of graphs employed for both training and testing (denoted as 'Train' and 'Test' respectively).

|         | IMDB  | COLLAB | TWITTER | RB     | SPECIAL | RB200  | RB500  | ER/BA/WS |
|---------|-------|--------|---------|--------|---------|--------|--------|----------|
| Nodes   | 19.77 | 74.49  | 131.76  | 216.67 | 106.89  | 197.28 | 540.79 | 158.10   |
| Density | 0.51  | 0.52   | 0.205   | 0.218  | 0.530   | 0.205  | 0.179  | 0.105    |
| Train   | 800   | 600    | 777     | 200    | 160     | 400    | 400    | 400      |
| Test    | 200   | 1000   | 196     | 400    | 40      | 100    | 100    | 100      |

# D   Additional Experiments on MIS

Besides the datasets of Table 1, we also performed experiments in the IMDB dataset and in three known graph distributions: *Erdos Renyi*, *Barabasi Albert* and *Watts Strogatz*. Table 5 shows that all methods, except for the random comparator, always achieve optimal results for the IMDB dataset, since the size of the IMDB instances is extremely small, as depicted by Table 4. Moreover, it is worth noticing that, on the ER dataset, our method outperforms the greedy heuristic in the mixed roll-outs case.

Table 5: Test set approximation ratios (higher is better) on one dataset made with real-world instances (IMDB) and three datasets made by three known graph distributions (ER/BA/WS). We report the average approximation ratio along with the standard deviation.

|                                  | IMDB              | ER                | BA                | WS                |
|----------------------------------|-------------------|-------------------|-------------------|-------------------|
| Comparator (Normal Roll-outs)    | 1.000             | $0.930 \pm 0.068$ | $0.922 \pm 0.089$ | $0.823 \pm 0.119$ |
| Comparator (Mixed Roll-outs)     | 1.000             | $0.954 \pm 0.058$ | $0.942 \pm 0.055$ | $0.831 \pm 0.116$ |
| Greedy MIS                       | 1.000             | $0.950 \pm 0.045$ | $0.959 \pm 0.045$ | $0.937 \pm 0.058$ |
| Random Comparator                | $0.874 \pm 0.261$ | $0.584 \pm 0.250$ | $0.490 \pm 0.251$ | $0.610 \pm 0.185$ |
| Simple Local Search (10s)        | 1.000             | $0.670 \pm 0.184$ | $0.533 \pm 0.189$ | $0.707 \pm 0.141$ |
| SCIP 8.0.3 (1s)                  | 1.000             | $0.908 \pm 0.147$ | $0.919 \pm 0.193$ | $0.926 \pm 0.150$ |
| SCIP 8.0.3 (5s)                  | 1.000             | $0.944 \pm 0.067$ | $0.969 \pm 0.044$ | $0.977 \pm 0.054$ |
| Gurobi 10.0 (0.5s)               | 1.000             | $0.945 \pm 0.074$ | $0.916 \pm 0.128$ | $0.990 \pm 0.030$ |
| Gurobi 10.0 (1s)                 | 1.000             | $0.960 \pm 0.061$ | $0.956 \pm 0.101$ | $0.997 \pm 0.014$ |
| Gurobi 10.0 (5s)                 | 1.000             | $0.988 \pm 0.026$ | $0.998 \pm 0.014$ | $0.999 \pm 0.004$ |

# E   Experiments on Minimum Vertex Cover

Experiments for the MVC problem are performed over the RB200 and RB500 Xu et al. [2007], which were specifically designed to generate hard instances. In addition, we evaluate our results by comparing them with the methodologies proposed in *Erdos' GNN* Karalias and Loukas [2020], *RUN-CSP* Toenshoff et al. [2019], and *Meta-EGN* Wang and Li [2023]. Specifically, the results presented in Table 6 are sourced from Wang and Li [2023], with the exception of the comparator and Greedy MVC, and all experiments were conducted using the same datasets as in their study.[Wang and Li, 2023].

Among the neural approaches, Table 6 highlights similar results for the two datasets, and our model achieves the best result for the RB500 dataset. Moreover, as depicted by the last rows of Table 6, optimal solvers do not always reach the optimal value over the graphs of the datasets, since, as mentioned before, RB instances are known to be hard.

Our model, benefiting from dynamic programming techniques, possesses the ability to decompose complex instances into more manageable sub-instances. As a result, it shows good skills in addressing the inherent complexity of the RB200 and RB500 datasets, surpassing other methods in this regard.

Table 6: Test set approximation ratio (lower is better) for different methods on real-world datasets. We present the performance of different algorithms on the minimum vertex cover problem (MVC). For each cell in the table, the average approximation ratio is reported together with the standard deviation. The time limit of solvers is reported next to the name. If the standard deviation is not reported, then the algorithm achieves always the optimal solution over all graphs of the dataset. The best performances are reported in bold.

| Method ($\downarrow$) Dataset ($\rightarrow$) | RB200 | RB500 |
|---|---|---|
| CMP | $1.031 \pm 0.006$ | $\mathbf{1.015 \pm 0.004}$ |
| Erdos' GNN | $1.031 \pm 0.004$ | $1.021 \pm 0.002$ |
| Meta-EGN | $\mathbf{1.028 \pm 0.005}$ | $1.016 \pm 0.002$ |
| RUN-CSP | $1.124 \pm 0.001$ | $1.062 \pm 0.005$ |
| Greedy MVC | $1.027 \pm 0.007$ | $1.014 \pm 0.003$ |
| Random CMP | $1.063 \pm 0.027$ | $1.031 \pm 0.015$ |
| SCIP 8.0.3 (1s) | $1.017 \pm 0.104$ | $1.025 \pm 0.018$ |
| SCIP 8.0.3 (5s) | $1.016 \pm 0.007$ | $1.011 \pm 0.003$ |
| Gurobi 10.0 (0.5s) | $1.009 \pm 0.006$ | $1.013 \pm 0.004$ |
| Gurobi 10.0 (1s) | $1.002 \pm 0.003$ | $1.012 \pm 0.003$ |
| Gurobi 10.0 (5s) | $1.000 \pm 0.001$ | $1.004 \pm 0.003$ |

# F  Additional Background Information

In our work, we use Gurobi as a matter of comparison for our results. Since Gurobi is an Integer Linear Programming (ILP) solver, we believe it is useful to briefly revise the ILP formulations of the MIS and MVC problems. In particular, given a graph $G(V, E)$ with $n = |V|$, we use binary decision variables $X = (x_i)_{1 \leq i \leq n} \in \{0, 1\}^n$ for each vertex $i \in V$ such that $x_i = 1$ if vertex $i$ is considered as part of the solution and $x_i = 0$ otherwise. In ILP, the goal is to assign values to every $x_i$ variable such that a function $F(G, X)$, called energy function, is either maximized or minimized under a set of constraints.

## F.1  Integer Linear Programming for the Maximum Independent Set

In MIS, the energy function is defined as the sum of every binary decision variable: $F(G, X) = \sum_{i \in V} x_i$. Since every edge $(i, j)$ can have at most one of its nodes in the independent set (otherwise the independent condition is violated), the sum of decision variables of $i$ and $j$ is at most one:

$$
\begin{aligned}
\text{maximize} \quad & \sum_{i \in V} x_i \\
\text{subject to} \quad & x_i + x_j \leq 1 \quad \forall (i, j) \in E \\
\text{and} \quad & x_i \in \{0, 1\} \quad \forall i \in V \ .
\end{aligned}
\tag{3}
$$

## F.2  Integer Linear Programming for the Minimum Vertex Cover

Differently from MIS, the ILP formulation for the MVC consists in minimizing the objective function $F(G, X) = \sum_{i \in V} x_i$. Since every edge of the graph has to have at least one of the two nodes in the vertex cover set (otherwise the edge wouldn't be covered), the sum of the decision variables of the two nodes defining the edge has to be greater (or equal) to one:

$$
\begin{aligned}
\text{minimize} \quad & \sum_{i \in V} x_i \\
\text{subject to} \quad & x_i + x_j \geq 1 \quad \forall (i, j) \in E \\
\text{and} \quad & x_i \in \{0, 1\} \quad \forall i \in V \ .
\end{aligned}
\tag{4}
$$

# G Ablation Study

In this section, we perform an ablation study on several parameters of the comparator function (without using mixed roll-outs) on the COLLAB dataset. The following parameters will be tweaked:

- $D$: The dimensionality of the initial node embeddings, and the output size of the layers in the network.

- $K$: The number of iterations in the Graph Embedding Module.

- $L$: The number of fully connected layers in the network.

We experiment linearly with the following parameters: $D \in [8, 16, 32, 48, 64]$, $K \in [1, 2, 3, 4, 5]$, and $L \in [2, 3, 4, 5]$, with a base configuration of $D = 32$, $K = 3$, and $L = 4$. This means that when testing with $D = 48$, the model configuration will be $D = 48$, $K = 3$, and $L = 4$. The meaning of these parameters can be found in greater detail in Sec. 4.2.

All models were trained for 200 epochs, with the experiments performed on a single GPU with 6GB RAM.

Table 7: Model Performance for a different value of $D$. The best performance is highlighted in bold.

| Parameter (D) | 8 | 16 | 32 | 48 | 64 |
|---|---|---|---|---|---|
| Model Performance | $0.976 \pm 0.053$ | $0.972 \pm 0.050$ | $\mathbf{0.990 \pm 0.049}$ | $0.980 \pm 0.055$ | $0.984 \pm 0.048$ |

Table 8: Model Performance for a different value of $K$. The best performance is highlighted in bold.

| Parameter (K) | 1 | 2 | 3 | 4 | 5 |
|---|---|---|---|---|---|
| Model Performance | $0.912 \pm 0.097$ | $0.944 \pm 0.068$ | $0.990 \pm 0.049$ | $\mathbf{0.992 \pm 0.037}$ | $0.960 \pm 0.059$ |

Table 9: Model Performance for a different value of $L$. The best performance is highlighted in bold.

| Parameter (L) | 2 | 3 | 4 | 5 |
|---|---|---|---|---|
| Model Performance | $0.922 \pm 0.084$ | $0.987 \pm 0.057$ | $\mathbf{0.990 \pm 0.049}$ | $0.993 \pm 0.046$ |

From Tables 7 to 9, we can see that a value of $D = 32$, $K = 3$, and $L = 4$ is a reasonable choice for the model architecture, also taking into account runtimes during inference for a model with smaller parameters. Interestingly, the model performance starts to degrade for $K = 5$, possibly due to over-smoothing.

# H Broader Impact

Designing neural networks for combinatorial optimization (CO) problems is still in its infancy, and as such there are a lot of exciting questions. Nonetheless, delving even more in this direction might potentially impact our society. Important fields like medicine or biology require the solution of CO problems in a short time since the execution time can be critical for the health of a patient. Therefore, tackling CO problems with deep learning techniques can bring important benefits to society, but can also be used for malicious purposes. Therefore, we do encourage the community to consider those challenges and perspectives.

# I Omitted Proofs

**Theorem 1**. *Let a graph $G(V, E) \in \mathcal{G}$. Then for any vertex $v \in V$ with $d(v) \geq 1$,*

$$|\text{MIS}(G)| = \max\left(|\text{MIS}\left(G/\mathcal{N}(v)\right)|, |\text{MIS}(G/\{v\})|\right) .$$

*Proof.* Let $G(V, E)$ be a graph and $\mathrm{MIS}(G)$ be its maximum independent set. We want to show that for any vertex $v \in V$ with $d(v) \geq 1$, the size of $\mathrm{MIS}(G)$ can be obtained by either removing $v$ or removing its neighbors $\mathcal{N}(v)$.

Consider two cases:

- $v \in \mathrm{MIS}(G)$

  In this case, if we remove $\mathcal{N}(v)$ from $G$, the resulting graph is denoted as $G' = G \setminus \mathcal{N}(v)$. Since the neighbors of $v$ cannot be in the maximum independent set, removing it does not affect the size of $\mathrm{MIS}(G)$. Therefore, $\mathrm{MIS}(G) = \mathrm{MIS}(G')$.

- $v \notin \mathrm{MIS}(G)$

  In this case, we can remove $v$ from $G$ to obtain the graph $G' = G \setminus \{v\}$. Since $v$ is not in the maximum independent set, removing it does not affect the size of $\mathrm{MIS}(G)$. Therefore, $\mathrm{MIS}(G) = \mathrm{MIS}(G')$.

By considering these two cases, we have shown that for any vertex $v \in V$, the maximum independent set $\mathrm{MIS}(G)$ can be obtained by either removing $v$ or removing its neighbors $\mathcal{N}(v)$. Thus, we can express the size of the maximum independent set as follows:

$$|\mathrm{MIS}(G)| = \max\left(|\mathrm{MIS}\left(G/\mathcal{N}(v)\right)|, |\mathrm{MIS}(G/\{v\})|\right) .$$

$\square$

**Theorem 2**. *Let a consistent comparator* $\mathrm{CMP} : \mathcal{G} \times \mathcal{G} \mapsto \{0, 1\}$. *Then the algorithm* $\mathcal{A}^{\mathrm{CMP}}(\cdot)$ *always computes a Maximum Independent Set,* $\mathbb{E}\left[\left|\mathcal{A}^{\mathrm{CMP}}(G)\right|\right] = |\mathrm{MIS}(G)|$ *for all* $G \in \mathcal{G}$.

*Proof.* Let us consider a consistent comparator $\mathrm{CMP}$. We will establish Theorem 2 with an induction on the number of edges $i$.

- *Induction Basis ($i = 0$):* Let $G(V, E) \in \mathcal{G}[0]$ then $\mathcal{A}^{\mathrm{CMP}}(G) = V = \mathrm{MIS}(G)$ .

- *Induction Hypothesis*: $\mathbb{E}\left[\left|\mathcal{A}^{\mathrm{CMP}}(G)\right|\right] = |\mathrm{MIS}(G)|$ for all $G \in \mathcal{G}[j]$ with $j \leq i$ .

- *Induction Step*: Let $G \in \mathcal{G}[i+1]$ and consider a node $v$ with degree $d(v) \geq 1$. Consider also the graphs $G_0 := G/\{v\}$ and $G_1 := G/\{\mathcal{N}(v)\}$. Both $G_0$ and $G_1$ admit less than $i$ edges and thus by the inductive hypothesis, $\mathbb{E}\left[\left|\mathcal{A}^{\mathrm{CMP}}(G_0)\right|\right] = |\mathrm{MIS}(G_0)|$ and $\mathbb{E}\left[\left|\mathcal{A}^{\mathrm{CMP}}(G_1)\right|\right] = |\mathrm{MIS}(G_1)|$. Hence $\mathrm{CMP}(G_0, G_1) = 0$ if and only if $|\mathrm{MIS}(G_0)| \geq |\mathrm{MIS}(G_1)|$. As a result, $|\mathcal{A}_{\mathrm{CMP}}(G)| = \max\left(|\mathrm{MIS}(G_0)|, |\mathrm{MIS}(G_1)|\right)$ and Theorem 1 implies that $|\mathcal{A}_{\mathrm{CMP}}(G)| = |\mathrm{MIS}(G)|$ .

$\square$

## J   Additional Experiments on Consistency

In an attempt to gain insight into the decisions of the comparator, we measure something that we refer to as the **consistency** value of a comparator. The consistency of a comparator is the proportion of graph pairs in which the equation of consistency from definition 3 is fulfilled.

Fig. 3 shows the consistency values as the training progresses for three different benchmark datasets. Overall, the consistency curves have an increasing behavior, indicating an increase in the consistency of the decision that a comparator makes as training progresses.

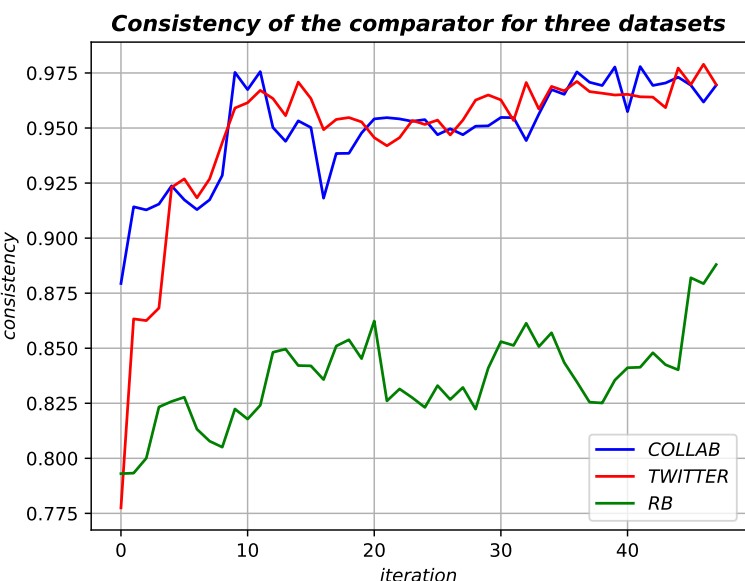

Figure 3: The figure highlights the behavior of the consistency of the comparator for the three datasets: COLLAB, TWITTER, and RB. The consistency (y-axis) represents the confidence of the comparator during the generation of the recursive tree. A consistency value close to 1 indicates high confidence in the comparator. Moreover, the consistency is calculated during the training process at the end of every iteration (x-axis), where an iteration corresponds to a total of 10 epochs. The consistency value associated with iteration 0 is obtained at the beginning of iteration 1 and indicates the consistency of the random comparator.

