# OpenReview forum: "Maximum Independent Set: Self-Training through Dynamic Programming"
_NeurIPS.cc/2023/Conference — NeurIPS 2023 poster_

### Official Review · Reviewer_v2kw · 2023-07-05

**Soundness:** 3 good
**Presentation:** 3 good
**Contribution:** 3 good
**Rating:** 6
**Confidence:** 5

**Summary:**

Maximum Independent Set:
A Dynamic programming approach (DP) to GNNs. The idea is to solve a Combinatorial Optimization problem, in particular the Maximum Independent Set (MIS): max set of nodes so that no two nodes are neighbors (set of nodes not linked by edges). The problem is NP-Hard but a GNN can generate an approximation. DP partitions the graph in subgraphs. The main idea is to avoid the exponential space.

Theorem 1 enables the recursive search. The idea is to remove nodes recursively or their neighbors. Basically, if a vertex is in the MIS, its neighbors cannot be!

GNNs for MIS. The approach is incremental.
A) Build comparator CMP based on theorem 1, for determining whether to graphs have approximately MIS size.
B)  Implement CMP using GNNs. Initially, we have neighbors and anti-neighbors (as the degree of freedom). However, the initial embedding is ZERO for ALL of them. The new embedding for a node comes by concatenating that of its neighbors with that of the anti-neighbors. Later on, when this information is pooled to form a global embedding and fed to an MLP, its decision will be sent backward. This is explained in section 5.

Self-Learning using CMP. Since the problem is NP-Hard it is reasonable to use the randomized CMP as a provider of examples. In this regard, Theorem 2 guarantees that any consistent graph comparing function CMP (the expectation of the randomized CMP is greater for the graph it has to be) induces an optimal algorithm ACMP for MIS. The basic idea is to choose to parameterize  CMP_theta in such a way that it is consistent. This is exactly the GOAL OF TRAINING.

Experiments. Compare the proposal with greedy methods. Usually greedy methods works very well but in some graphs. The method seems to generalize a bit.

I agree with the conclusions: “We firmly believe that a thorough investigation into the interplay between Dynamic Programming and self-training techniques can pave the way for
new deep-learning-oriented approaches for demanding CO problems”.

**Strengths:**

A nice paper for illustrating the use of GNNs in CO problems through self learning, specially in NP-Hard Problems.
Nice time complexity for a fixed budget.

**Weaknesses:**

Despite the framework is nice, there is no insight in terms of verifying whether the resulting embeddings of the nodes verify the decision taken. A probable approach is to hash the compatible nodes to one code and the incompatible one to another code. In this way the method will be more explanatory and will enlarge the consistency of the response.



**Questions:**

How do the parameters of the CMP_theta evolve? How far/close are them of resturning consistent/inconsistent responses? In other words, use the analysis of these parameters to diagnose how hard is the problem for different graphs.

**Limitations:**

Maybe better GNNs could improve the results. Agree with the authors the lack of convergence analysis. In this regard, it seems to me that making the CMP_theta more explainable will help.

---

> ### Author Rebuttal · Authors · 2023-08-09
>
> We thank the reviewer v2kw for their feedback. We address their concerns below:
>
> > Q: “How do the parameters of the $CMP_{\theta}$ evolve? How far/close are they returning consistent/inconsistent responses? In other words, use the analysis of these parameters to diagnose how hard the problem is for different graphs.”
>
> We are thankful to the reviewer for the suggestion. We analyze the GNN decisions by calculating the consistency value which is the percentage of the graphs pairs at which the equation of Definiton 3 holds (consistency). Fig. 1 in the pdf shows the consistency values as training proceeds. Overall, the **consistency curves have an increasing behavior, indicating an increase in the comparator consistency as training goes on**.
>
>
> We are happy to address any other concerns the reviewer v2kw might have.

---

> > ### Author Response · Authors · 2023-08-15
> > **Are there additional comments from Reviewer v2kw?**
> >
> > Dear reviewer v2kw,
> >
> > we hope that our response covers the key questions of consistency from the original review. We are **open to your suggestions**, since the summary indicates a clear understanding of our work. If you have any additional comments or questions, we are happy to address them.
> >
> > Best regards,
> >
> > Authors

---

> > > ### Comment · Reviewer_v2kw · 2023-08-18
> > > **Response to rebuttal**
> > >
> > > Dear authors,
> > > Im happy with your response to the parameters issue. However, my concern wrt explainability
> > >
> > > "Despite the framework is nice, there is no insight in terms of verifying whether the resulting embeddings of the nodes verify the decision taken. A probable approach is to hash the compatible nodes to one code and the incompatible one to another code. In this way the method will be more explanatory and will enlarge the consistency of the response"
> > >
> > >  is not addressed by the authors.
> > > Thanks a lot!

---

> > > > ### Author Response · Authors · 2023-08-19
> > > > **Question to Reviewer v2kw for the suggested evaluation steps**
> > > >
> > > > Dear Reviewer v2kw,
> > > >
> > > > In our initial response we provided the additional plots quantifying the consistency of our self-trained model. As these figures suggest, the models become more consistent over time which we believe is indicative in explaining the choices of the model.
> > > >
> > > > We are **happy to include additional evaluations** in case the previous experiment does not address your comment. However, in this case **we would need some further clarifications**. Your comment suggests that for every (random) node we are at, to separate the nodes into compatible (i.e. everything apart from its neighbors) and non-compatible (i.e., its neighbors). However we are not certain of the next steps beyond that point.
> > > >
> > > > Looking forward for the next steps.

---

### Official Review · Reviewer_T1ds · 2023-07-06

**Soundness:** 2 fair
**Presentation:** 3 good
**Contribution:** 3 good
**Rating:** 6
**Confidence:** 1

**Summary:**

This paper studies the combination optimization problem, maximum independent set, via self-training. By training a consistent graph comparator function to determine the larger MIS of two different graphs, the MIS of original graph can be obtained recursively efficiently.

**Strengths:**

1. The notations and presentation are clear.
2. The idea of using the intermediate results during algorithm execution to supervise the model itself is novel.

**Weaknesses:**

1. The most important component in the proposed method is the consistent graph comparator. It would be better to include more detailed explanation on why the learned graph comparator will be consistent.
2. As the algorithm $A$ depends on $CMP_\theta$, and the whole training pipeline relies on $A$, what will happen if $CMP_\theta$ makes a lot of mistakes initially? This should be discussed.
3. The computational complexity/overhead should be discussed.

**Questions:**

See in weaknesses.

**Limitations:**

Limitations are mentioned. Negative societal impact is not applicable.

---

> ### Author Rebuttal · Authors · 2023-08-09
>
> We thank the reviewer T1ds for their feedback. We address their concerns below:
>
> > Q: Include a more detailed explanation on why the learned graph comparator will be consistent.
> 	Notice that the optimization problem of Eq. 2 effectively asks for more and more consistent comparators.
>
> Nevertheless, to provide an empirical demonstration, we opted to conduct the following experiment. We are thankful to the reviewer for the suggestion. We analyze the GNN decisions by calculating the consistency value which is the percentage of the graphs pairs at which the equation of Definiton 3 holds (consistency). Fig. 1 in the pdf shows the consistency values as training proceeds. Overall, the **consistency curves have an increasing behavior, indicating an increase in the comparator consistency as training goes on**.
>
>
> We are happy to address any other concerns the reviewer T1ds might have.

---

> > ### Author Response · Authors · 2023-08-20
> > **Are there any last concerns from Reviewer T1ds?**
> >
> > We are happy to address any last concerns the reviewer T1ds might have. Beyond the consistency, let us further clarify some of the points raised by the reviewer:
> >
> > 2. Naturally, the untrained comparator makes a lot of mistakes in the beginning. These mistakes are also indicated in the poor performance of the untrained comparator presented in Table 1 (in the paper). However, through additional roll-outs in the produced sub-graphs during the execution of Algorithm~1, the comparator is able to realize its past mistakes and update the weights that lead to fewer and fewer mistakes. We believe that the latter is nicely depicted in the additional plots that we have provided depicting the increasing behavior of the comparator (see Figure in the pdf). *We remark that being more consistent implies fewer mistakes*. In the revised version of our work we will incorporate the above valuable discussion.
> >
> > 3. We are thankful to the reviewer for raising this point. The complexity of our inference algorithm is in the worst case $O(n^3)$. In the revised version of our work we will explicitly mention the above (see also our response [b8RC](https://openreview.net/forum?id=igE3Zbxvws&noteId=zIK0d3OVCR) and the response to  [qbTV](https://openreview.net/forum?id=igE3Zbxvws&noteId=LaaSZFxI4a)).
> >
> > If the reviewer has any remaining concerns, we would be happy to address them.

---

> ### Comment · Reviewer_T1ds · 2023-08-20
> **response**
>
> Thank the authors' explanation. I raise my score from 5 to 6.

---

> > ### Author Response · Authors · 2023-08-20
> > **Thankful to Reviewer T1ds for raising their score**
> >
> > Dear Reviewer T1ds,
> >
> > we are thankful for recognizing the effort for our explanations and increasing the score.
> >
> > Best regards,
> >
> > Authors

---

### Official Review · Reviewer_b8RC · 2023-07-07

**Soundness:** 3 good
**Presentation:** 4 excellent
**Contribution:** 3 good
**Rating:** 6
**Confidence:** 4

**Summary:**

This work uses a dynamic programming framework for approximately solving the maximum independent set (MIS) and minimum vertex cover (MVC) problems. A graph neural network (GNN) is used to replace a heuristic step in the randomized version of the DP algorithm. Essentially, MIS (and MVC) can be broken down into (2^n) recursive decisions about whether to include a vertex in the output set or not. The randomized version doesn’t go through all of these, but decides for a random vertex whether or not to include it. This setup ensures that the output will be a valid independent set regardless of the decisions. In this paper, the decisions are made by a trained GNN (heuristic). The GNN is trained by comparing the output IS with random independent sets formed by performing rollouts starting with the alternative choice. E.g., if the GNN decides to include vertex v in the IS, then the rollouts are performed on the graph without v. The greedy rollout is included in the mixed version, to potentially improve this training signal. The GNN architecture used is referred to as GEM, but as far as I can tell, the GNN is exchangeable for any other GNN architecture.

**Strengths:**

1. This is a neat setup for DP problems on graphs. The framework ensures that the outputs are valid, and using a GNN as a heuristic can make sense. It's a nice idea to replace a heuristic step in an algorithm with a GNN (but this is not the first work to do so).
2. The approach is clearly laid out and easy to follow.
3. Some of the results are promising. The GNN also clearly outperforms the random CMP, indicating that the GNN is learning something useful and contributing positively to the algorithm.

**Weaknesses:**

1. It is not clear how much impact this paper might have. I’m sure the framework can be used for some other CO problems where the output is a subset of the input, but it is not clear how widely applicable it might be, especially given its runtime complexity.
2. The main novelty seems to be the use of a GNN for the decision step in the algorithm.
3. GEM is given a new name, but it is essentially fully-connected message passing, where neighbours and non-neighbours are aggregated separately. This is not a novel contribution, so I don’t think it deserves a new name.
4. The proposed model is not scalable to larger graphs. Already the GNN used (GEM) is n^2 per layer (n being the number of nodes), which simply doesn’t scale to large graphs. Moreover it is applied n times at inference, giving n^3 runtime. What are the largest graphs you can run on? How does the complexity compare with the complexity of the other approaches?
5. Greedy baseline outperforms the proposed approach on 2/4 datasets.
6. The results are not very convincing for MVC.

**Questions:**

1. Have you tried using any other GNN architectures (GIN, GAT, Erdos, Graph Transformer, ...) for the heuristic step?
2. Are the anti-neighbors needed? How is the performance without (this would help in lowering the overall complexity)?
3. How many rollouts are needed for the results?
4. Why not use a GNN to decide on the next vertex to consider rather than doing it randomly? There might be some vertices that are easier to decide whether to include at a given time.
5. Why is SPECIAL missing from Table 2?
6. Is RB or TWITTER really out-of-distribution for COLLAB? Maybe those datasets are very similar in distribution.
7. What about using only the greedy rollout?
8. Some symmetry breaking is probably needed for the greedy approach (e.g. if there are (neighboring) nodes with the lowest degree in the graph), so multiple rollouts could be done here and the max taken. Do you consider only a single (random) rollout? Or do you run greedy multiple times?
9. The random CMP could be replaced by a degree-weighted random CMP, i.e. selecting lower degree nodes with a higher probability, but not necessarily taking a lowest degree node. This would essentially make the greedy approach a little more robust to “special” graphs, maybe providing a better baseline.
10. Do you have any insights into the GNN decisions?

**Limitations:**

yes

---

> ### Author Rebuttal · Authors · 2023-08-09
>
> We thank the reviewer b8RC for their feedback. We address their concerns below:
>
> > W1: Unclear impact of the paper and applicability to other CO problems
>
> We agree with the reviewer that extending to other CO problems is not trivial, but we respectfully disagree on the significance of our work. The main motivation of our work is to investigate Dynamic Programming, a well-established technique, in combination with deep learning models. We believe that our results clearly indicate how fertile this coupling is and is of interest for the ‘learning to optimize’ audience of NeurIPS. This is also clearly identified by R. v2kw.
> ____
>
> > W2: No novelty on GEM
>
> We are not aware of any previous work using the latter aggregation mechanism. In case the reviewer can provide us with the specific reference we are happy to provide the due credits.
> ____
> > W3: Scalability to larger graphs
>
> The time complexity of our algorithm is $O(n^3)$ which indeed poses limitations on applying it to very large graphs . However we remark that it requires similar (or even better) running time of previous DNN approaches (see Table 1 and 2 where s/g stands for seconds per graph). We emphasize that this runtime is even obtained with code written on PyTorch modules and we firmly hold that further implementing our model with low-level libraries could further boost the runtime performance. For example, at each step, the adjacency matrix of the graph must be modified by removing either the selected node or its neighbors. This is an expensive operation when using NetworkX (which is a standard library). Furthermore, by batching graphs in order to allow for GPU-based parallelization, which is not currently done, we expect to significantly improve training and inference results of the model.
> ____
> > W4: Performance when compared to the greedy baseline
>
> Beating the greedy heuristic for MIS with deep learning methods that do not use any domain knowledge on the problem is a challenging task. We remark that the latter holds for all previous learning approaches out of which our method presents the best performance. We believe that the latter does neither lower the value of our work nor the value of the learning to optimize line of research, that it is a way more recent research direction with respect to the human-design heuristics.
> ____
>
> > W5: Performance on MVC
>
> The focus of this work is on MIS (which is already declared in the title of the paper). Nevertheless, we make an effort to demonstrate the extension of our work to other CO tasks, such as MVC. We do believe that with an improved architecture (e.g. as in the next question), we could further improve improvements in the MVC, but even our approach from MIS readily achieves a decent performance in MVC.
> ____
>
> > Q1: Can you use other GNN architectures?
>
> Yes, but we have not tried in detail, since the focus of this work is not the GNN architecture per se. We believe this is a future direction for our work, which we will explicitly mention in the final version.
> ____
>
> > Q2: Role of anti-neighbors
>
> We appreciate the recommendation from the reviewer and agree about the comment regarding the computational complexity. Inspired by this, we conducted the related experiment and verified that the complexity without the anti-neighbors is lower, but the performance deteriorates. The results in Table 1 (in the pdf) indicate how the performance without the anti-neighbors (AN) deteriorates. We will include the experiment in the revised version.
> ____
>
> > Q3: How many rollouts are needed for the results?
> Table 3 (in the pdf) reports the approximation ratio for different rollout values. Notice that the optimal rollouts for both COLLAB and TWITTER belong in the range of 5 or 10.
> ____
>
> > Q4: Why not use a GNN to decide on the next vertex to consider rather than doing it randomly? There might be some vertices that are easier to decide whether to include at a given time.
>
> Indeed, it is a reasonable tweak that we have not considered since we wanted to investigate the performance of the vanilla version of the DP-based learning approach. We will explicitly mention this in the final version as a future step.
> ____
>
> > Q5: Why is SPECIAL missing from Table 2?
>
> We notice that RB and SPECIAL distributions share common patterns. For instance, they have big clusters of fully-connected nodes, but clusters are almost independent. Therefore, this is adding redundant information.
> ____
> > Q6: Is RB or TWITTER really out-of-distribution for COLLAB?
>
> The graph distributions between the three datasets are quite different. Please check Table 2 in the pdf.
> ____
> > Q7: What about using only the greedy rollout?
>
> It achieves similar performance.
> ____
> > Q8: Do you run greedy multiple times or a single rollout?
>
> We run greedy multiple times exactly as the reviewer suggests.
> ____
>
> > Q9: Degree-weighted random CMP baseline
>
> We tried the recommendation of the reviewer, but, unfortunately, it did not produce a good result. The result of the pure random CMP over the SPECIAL graphs is 0.225 +- 0.279 while the result of the degree-weighted random CMP is 0.172 +- 0.192. Even if the chosen node is not the lowest degree one, the comparator has to make a choice. This choice is wrong 50% of the times, since the comparator is random, producing a low approximation ratio.
> ____
> > Q10: Insights into the GNN decisions
>
> We analyze the GNN decisions by calculating the consistency value which is the percentage of the graphs pairs at which the equation of Definiton 3 holds (consistency). Fig. 1 in the pdf shows the consistency values as training proceeds. Overall, the **consistency curves have an increasing behavior, indicating an increase in the comparator consistency as training goes on**.
>
> We are happy to address any other concerns the reviewer b8RC might have.

---

> > ### Author Response · Authors · 2023-08-15
> > **Are there any remaining questions from reviewer b8RC?**
> >
> > Dear reviewer b8RC,
> >
> > We appreciate your constructive feedback and questions. We are wondering whether there are any additional questions that we might be able to answer for the reviewer.
> >
> > The key questions so far concern both the *methodology*, e.g., requirement for the anti-neighbors, and the *experimental setup and results*, e.g., the the difference between the datasets (please check Table 2 in the provided single-page pdf for a detailed answer), or the insights for the GNN decisions.
> >
> > Please let us know if you have any remaining questions. We are happy to answer them. If you are satisfied with our responses, we would be grateful if you could re-evaluate your score.
> >
> > Best regards,
> >
> > Authors

---

> > > ### Comment · Reviewer_b8RC · 2023-08-16
> > >
> > > Thank you for addressing my comments and questions. I appreciate the effort that went into the rebuttal.
> > >
> > > **W2:** The authors may well be right here. I have not found a clear reference to fully connected message passing. However, the idea is similar to a graph transformer, particularly a graph transformer using edge features to indicate the existence of an edge. It is also equivalent to an R-GCN [1] with two edge types - edges and non-edges. I do not view it as a significant contribution, but neither do the authors present it as such.
> > >
> > > [1] Schlichtkrull, Michael, et al. "Modeling relational data with graph convolutional networks." The Semantic Web: 15th International Conference, ESWC 2018, Heraklion, Crete, Greece, June 3–7, 2018, Proceedings 15. Springer International Publishing, 2018.
> > >
> > > **W4 "Beating the greedy heuristic for MIS with deep learning methods that do not use any domain knowledge on the problem is a challenging task":**
> > >
> > > I don't necessarily agree with this statement. The overall method here certainly is using domain knowledge, it is designed around an algorithm for the problem. The deep learning method replaces one (key) step of the algorithm. I also do not agree that beating the greedy heuristic is always challenging. This depends on the context. In this context, the greedy heuristic is quite simple - to choose the lowest degree node. However, the task of the comparator is to decide whether to include a random node in the independent set, which does not necessarily make it easy to pick the lowest-degree nodes. So the task is certainly not trivial.
> > >
> > > **Q2:** Thank you for adding this experiment. The results are interesting. The gap is surprisingly large.
> > >
> > > **Q3:** Again, thank you for adding this.
> > >
> > > **Q6:** Thank you for the additional analyses.
> > >
> > > **Q7:** I don't see any results for this. But if this is the case, then I think the results should be added to the final version to make it clear that the greedy rollouts (in the mixed roll-out models) are the important ones and the normal rollouts could be dropped.
> > >
> > > **Q9:** How exactly was this done? Could you give more details?
> > >
> > > **Q10:** This is very interesting. Do you have the greedy baseline for these three datasets? (the greedy baseline would only use a node if the node is in a greedy solution)

---

> > > > ### Author Response · Authors · 2023-08-16
> > > > **Response to the questions of Reviewer b8RC**
> > > >
> > > > We thank the reviewer for their constructive comments and questions. We elaborate directly on the raised questions:
> > > >
> > > > **W2**: We are thankful for the reference, we will check and cite the corresponding paper. As the reviewer identifies, this is not a core claim, and indeed we believe follow-up works can further improve at this by using more tailored modules.
> > > >
> > > > ____
> > > > **W4**: We appreciate the rigorous remark by the reviewer. We agree that domain knowledge is used for the theoretical insights in our case. However, as the reviewer recognizes, we design an algorithm on the “learning to optimize” framework, which relies on deep networks. Besides, in the paper, we show the exact numbers of the greedy baseline, such that a reader is informed about the exact results of every method.
> > > > ____
> > > > **Q7**: Let us clarify further the previous response. Greedy roll-outs admit similar performance with mixed roll-outs in datasets at which the greedy heuristic performs well, such as in Twitter and Collab datasets. However, in datasets at which the greedy heuristic performs poorly, using just the greedy roll-outs during training leads to significantly worse performance than using normal or mixed roll-outs. In the table below, we present the performance of the various roll-outs in the case of SPECIAL graphs.
> > > >
> > > > | Dataset   | Normal CMP | Mixed CMP | Greedy CMP | Greedy Heuristic |
> > > > |-----------|----------------------|-----------------|-------------------|------------------------|
> > > > | SPECIAL   | $0.996 \pm 0.029$ | $0.994 \pm 0.035$ | $0.493 \pm 0.427$ | $0.131 \pm 0.055$ |
> > > >
> > > > Thus, using mixed roll-outs leads to robust performance even in case the greedy heuristic performs poorly.
> > > >
> > > > In the revised version of our paper we will incorporate and elaborate on these results. We thank the reviewer for their helpful suggestion.
> > > >
> > > > ____
> > > > **Q9**: We tested on initial graphs $G \sim \mathcal{D}\_\text{SPECIAL}$, where $\mathcal{D}\_\text{SPECIAL}$ is the dataset with counterexamples to the greedy baseline, the constructed Degree-Weighted Random CMP works as follows: At every recursive step as defined in Algorithm 1, we don’t sample a node at random, but we sample a node according to the degree distribution of $G$. More specifically, we order the nodes from the largest to the smallest, and create a line with positive constant slope from the largest degree to the smallest degree node. This line represents the probability of being selected. Thus, lower-degree nodes have a larger probability of being selected. For the rest, the model acts identical to the Random CMP.
> > > >
> > > >
> > > > ____
> > > > **Q10**: Thanks for the interesting suggestion. In the resulting table below we have added the consistency of the greedy baseline for RB, Twitter and Collab. More precisely, after samling a graph $G \sim \mathcal{D}$, we compute an independent set given by the greedy solution. Then, for each node $u$ belonging to the greedy solution, we create the graph $G/\{u\}$ and $G/\{N(u)\}$. In case $Greedy(G/\{u\}) < Greedy(G/\{N(u)\})$, we count for a match (respectively for a mismatch) and output the resulting probability of a match.
> > > >
> > > > If the reviewer would like any further info on greedy, we would be glad to address them.
> > > >
> > > > | Dataset  | Greedy  |
> > > > |----------|---------|
> > > > | TWITTER  | $0.991 \pm 0.096$ |
> > > > | COLLAB   | $0.995 \pm 0.068$|
> > > > | RB       |   $0.868 \pm 0.339$    |
> > > > | SPECIAL | $0.394 \pm 0.489$ |
> > > >
> > > >
> > > > If all the concerns of the reviewer are addressed, we would be grateful if the reviewer reconsiders their score.

---

> > > > > ### Comment · Reviewer_b8RC · 2023-08-18
> > > > >
> > > > > **Q10:** This is the analysis I was looking for. Taken together with Figure 1, it seems that the learned comparator does not exceed greedy performance (in terms of consistency) for the 3 real-world graphs. I assume this would be different for SPECIAL, but this dataset is not included in Figure 1. Along with the results in Table 1, this suggests that the GNN might not learn anything much more complex than selecting low-degree nodes on real-world datasets (Twitter, Colab, RB). This could be further explored by checking the consistency between Greedy and the learned comparator (ignoring the optimum). On the other hand, clearly, the comparator learns something different on the SPECIAL dataset. This would also be interesting to analyse.
> > > > >
> > > > > Overall, thank you for the additional experiments and the thorough responses. I am happy to raise my initial score.

---

> > > > > > ### Author Response · Authors · 2023-08-20
> > > > > > **Response to the question of Reviewer b8RC**
> > > > > >
> > > > > > Dear Reviewer b8RC,
> > > > > >
> > > > > > We are thankful for your constructive suggestions that have helped us obtain additional insights from our framework.
> > > > > >
> > > > > > Following your suggestion, we analyze the consistency property during training on the SPECIAL dataset and show the results in the table below for 100 epochs (or 10 iterations equivalently). We observe that the consistency of the model exhibits a similar performance as the consistency of the COLLAB and TWITTER dataset in Figure 1 of the single-page PDF, exceeding greedy performance as the reviewer suggests. We provide the table since we cannot include an external link with the plot (according to the NeurIPS guidelines). Nevertheless, all of those results will be included in the final version.
> > > > > >
> > > > > > We thank the reviewer again for their insightful comments and for increasing the score.
> > > > > >
> > > > > > | Epoch       | 0     | 10    | 20    | 30    | 40    | 50    | 60    | 70    | 80    | 90    | 100   |
> > > > > > |-------------|-------|-------|-------|-------|-------|-------|-------|-------|-------|-------|-------|
> > > > > > | Consistency | 0.405 | 0.660 | 0.723 | 0.803 | 0.750 | 0.757 | 0.818 | 0.877 | 0.893 | 0.849 | 0.931 |

---

### Official Review · Reviewer_YVwa · 2023-07-07

**Soundness:** 3 good
**Presentation:** 4 excellent
**Contribution:** 3 good
**Rating:** 7
**Confidence:** 4

**Summary:**

This paper proposes a method to solve the maximum independent set (MIS) problem using self-trained dynamic programming and carefully-designed GNNs. The MIS problem is decomposed into dynamic programs and solved by comparing two reduced graphs.

**Strengths:**


It is interesting and innovative to address the MIS problem using GNNs in a self-training manner.
The proposed method is effective and efficient for solving the MIS problem.

**Weaknesses:**


The following improvements could enhance the experiments:

1. Baseline comparisons: In Line 74, several recent baselines are mentioned. It would be beneficial to include comparisons with these methods in the experiments to demonstrate the superiority of the proposed approach.

2. Evaluations on large-scale graphs: Existing methods [Schuetz et al., 2022b, Wang and Li, 2023] have demonstrated their effectiveness on larger-scale graphs with more than $10^4$ nodes. It would be valuable to showcase the results of the proposed method on such graphs.

3. Comparisons on model training: Since the proposed method employs self-training, it would be informative to compare the training cost of the proposed method with that of the baselines. Including such comparisons would provide insights into the efficiency of the proposed approach.

**Questions:**

None

**Limitations:**

Please check the weaknesses.

---

> ### Author Rebuttal · Authors · 2023-08-09
>
> We thank the reviewer YVwa for their feedback. We address their concerns below:
>
>
> > Q1: Baseline comparisons
>
> In Line 74, we present the various DNN approaches for various CO settings and adapting those to MIS might not be trivial. In our experimental evaluations we compare with the existing approaches for MIS. If the reviewer believes there is a relevant baseline that we should include in the comparisons, we are happy to consider it.
> ____
>
> > Q2: Evaluations on large-scale graphs
>
>
> We are thankful to the reviewer for the remark and the references. Let us clarify why the presented methods are not directly comparable with ours and the other deep-learning approaches that we compare against.
>
> Firstly, we remark that [Schuetz et al., 2022; Wang and Li, 2023] present only results for $d$-regular graphs for some small values of $d$ (3 and 5). Despite the fact that these graphs admit a large number of nodes, the number of edges scales only linearly with $n$ which is the reason why their methods can scale. This is a different setting to ours, where we do not make such assumptions about the graph distribution. Furthermore, notice that both [Schuetz et al., 2022; Wang and Li, 2023] use GNNs meaning that the inference algorithm admits at least $O(n^2)$ complexity for dense graphs.
>
> What is more, let us clarify why the learning of these approaches also differs from the aforementioned deep-learning approaches. [Schuetz et al., 2022] optimize the parameters of the GNN providing a fractional MIS-solution and then use a standard randomized rounding to convert to an integral one.  However, they *train* the GNN for each new graph separately during inference (or in other words, the parameter optimization of the GNN is part of the inference algorithm). The latter pipeline is very similar to the classical *relax and round* approach that has been used in the approximation algorithms for years (see section 4,5 and 6 in [1]). On the other hand [Wang and Li, 2023] do not derive a new learning algorithm but rather use their approach to tune the classical *greedy algorithm* whose practical favorable performance has been identified over the years.
>
> [1] The Design of Approximation Algorithms, Williamson 2009
>
> ____
>
> > Q3: Model training timing
>
> We are thankful for the recommendation by the reviewer. We ran the training process on the TWITTER dataset for 1000 iterations. Concretely, given the constructed dataset, we measured the time it required for forward + backward passes with our model and the EGN model from [1]. The results in the table below showcase that our method is quicker for this dataset.
>
> | Model | Twitter (seconds/graph) |
> |-------|-------------------------|
> | Ours | 0.45 |
> | EGN | 1.62 |
>
> Although the TWITTER dataset does not contain graphs with thousands of nodes, our method should scale with a similar rate to that of prior deep learning approaches for dense graphs.
>
>
> We are happy to address any other concerns the reviewer YVwa might have.

---

> > ### Comment · Reviewer_YVwa · 2023-08-17
> >
> > Thank you for your response, and my questions have been adequately addressed. At this point, I intend to maintain my current score.

---

> > > ### Author Response · Authors · 2023-08-17
> > > **We appreciate your feedback and questions**
> > >
> > > Dear Reviewer YVwa,
> > >
> > > we are thankful for your feedback and strong support for our work. We welcome any additional recommendations to improve our work.
> > >
> > > Best regards,
> > >
> > > Authors

---

### Official Review · Reviewer_qbTV · 2023-07-24

**Soundness:** 2 fair
**Presentation:** 3 good
**Contribution:** 3 good
**Rating:** 5
**Confidence:** 4

**Summary:**

The paper proposes a GNN based method to approximate the largest independent set for a graph. The algorithm proposes to train a GNN layer to decide at each iteration whether a randomly selected node should be a part of the independent set, the resulting graph (either losing the neighbours of the chosen node or only the node the itself) is then the input at the next iteration. For graph distribution datasets are used for empirical evaluation.

**Strengths:**

The paper employs a DNN module at an appropriate place, i.e. as the graph comparator function, which is sensible. This would be a reasonable contribution if the weaknesses would be addressed.

**Weaknesses:**

EDIT: Summary: Some of the weaknesses mentioned were incorrect (W1,W2,W4) (apologies to the authors). Some of the weaknesses simply need to be discussed in the paper (W3,W5,W6,W7,W9). In my opinion the deciding factor should be W8, does the AC think that simply using benchmarks with no real-world connection to the problem we are trying solve are good empirical evaluation (only used because prior work did and nobody checked that) is good enough. If Yes, the paper should clearly be accepted (score of 7), else it should be rejected as the accepted benchmark should be updated. Thus, I have updated my score to a 5, 7 if the authors decide to add a benchmark with a real world connection or explain why one of the current benchmarks has a real-world connection and thus represents the distribution of graphs one might find in practice.

1. I retract the first point in the bullet list below, it is indeed incorrect as the authors point out in the rebuttal.
2. In the second bullet point, I also did forget a word ("might") that does matter, however, I regarding remark 4 I maintain my criticism, while the interpretation that the authors provide in the rebuttal makes sense what is written in the paper is much more broad. I suggest they reword for the camera ready.
3. Regarding point 3, I thank the authors for addressing the computational complexity more directly, but the comparison in run-time to Gurobi seems unfair as that tries to find the optimal solution as far as I understood, which is naturally exponential in the worst case, so this seems to be an apple to oranges comparison, the greedy heuristic may be a better comparison point.
4. I also retract W4.
5. It is a weakness that the paper doesn't discuss this in it's current form.
6. I stand by W6 given the current phrasing in the paper which is "Since we observe unexpected performance from RUN-CSP on the COLLAB and RB269 datasets, we have omitted those results from the table." This doesn't explain why it has been removed, which is necesary.
7. Units should still be explained. One prior work doesn't make it standard enough to not mention it.
8. Prior work having used the benchmarks is in my opinion a weak argument without any other benchmarks. But I understand that this is a contentious issue within the ML community. I leave this up the AC, but as far as I can tell from the response of the authors, there is no particular use in the real-world to compute the MIS on something like the Twitter dataset.
9. I agree that such work is still valuable, but it's an obvious limitation that needs to be discussed and wasn't in the paper.

The paper has substantial weaknesses that prevent acceptance at this time in my view:

1. The authors demonstrated knowledge of what NP-hard means is poor, frequently mis-using the term and similarly the context of approximation algorithms is missing completely. In more detail: NP is a class of BINARY decision problems, i.e. the problem asks a yes or no question. Maximum Independent Set (Defintion 1) as defined in the problem is not such a problem and thus cannot be classified as NP-hard or anywhere in that complexity hierarchy. Leading to several incorrect claims in the paper (e.g. line 181). Furthermore, problems that are NP-hard are considered computationally infeasible for anything but the smallest instance sizes, so a sentence like "computationally infeasible for NP-hard problems" (line 21) does not make sense. If we want to discuss the optimisation problems associated with an NP-hard problem, e.g. finding the optimal route of the travelling salesman problem or finding the maximum independent set, this is a different computational complexity class with it's own hierarchy. This is especially important once we care about approximations, doubly so when we want to considered the randomized approximation algorithms, however, any such discussion is missing.
2. Various statements about the quality of the solutions of the algorithm are imprecise or plain wrong. For instance,  in line 136 the claim is made "there exist a reasonable graph comparing functions that i) are efficiently computable ii) lead to near optimal solutions" with no reference to any proof or evidence. These baseless claims decrease the authors credibility significantly. (another example can be found in remark 4 the second sentence).
3. The computational computational complexity of the algorithm is never discussed, which is particularly relevant given we are approximating solutions and thus there is a fundamental trade-off between time and the quality of the solution found. The algorithm proposed is of high-computational complexity, a single GEM module layer as proposed in equation 1. is already of complexity O(n^2) where n is the number of nodes. This would quickly be infeasible for problems of an interesting size and also raises the question why a GraphTransformer architecture wasn't considered. Furthermore, the GEM layers need to be applied O(n) time in the worst case, as the recursion depth of algorithm 1 is O(n) in the worst case. Thus giving us at least O(n^3) at inference time, neglecting the training time here, making the algorithm expensive to run. This is not compared against any other method either, e.g. the computational complexity of traditional solvers like Gurobi or other baseline methods is also not given.
4. Theorem 1 seems to be missing a +1, presumably when we remove a node v from the graph and add it to the independen set this should lead to a +1 appearing somewhere in the equation of |MIS(G)|.
5. The computational complexity of computing the expectation in the loss function given in equation 2 seems prohibitively high and there is no discussion about the variance in estimating the expectation by using only samples. In practice, this seems to be replaced with roll-outs another form of probabilistic estimation, whose complexity or variance remain unmentioned. Again, a claim is made that these roll-outs are better than the original expectation without any empirical or theoretical evidence.
6. The empirical evaluation removes the results from the RUN-CSP baseline with the words "unexpected performance" without any further justification, this is highly unusual and not acceptable practice.
7. Table 1 uses an unexplained unit of s/g.
8. The benchmark datasets aren't justified, why is it interesting to compute the MIS of a twitter graph? Given that the graph distribution will be highly relevant to performance this is important. While it is understandable to re-use benchmark datasets that prior work has used, it doesn't help the paper if the benchmark datasets aren't specfic to any real world problem people care about. At least one dataset needs to be of relevance to a real world problem, e.g. MIS is used in compiler optimisations.
9. There is no adequate discussion of running time or discussion of any benefits of the proposed method or DNN methods in general over classical optimisation solvers for the problem studied. Indeed, from Table 1. I cannot discern why a practioner would use the method proposed over Gurobi over SCIP.

**Questions:**

None.

**Limitations:**

No the limitations are not adressed, see the weaknesses.

---

> ### Author Rebuttal · Authors · 2023-08-09
>
> We thank the reviewer qbTV for their feedback. We address their concerns below:
>
> > Q1: MIS cannot be classified as NP-hard or anywhere in that complexity hierarchy
>
> We respectfully disagree with the reviewer. Maximum independent Set (MIS) is an NP-hard problem (e.g. see [D] published in ICML’20).
>
> Indeed MIS does not belong in NP since it is not a decision problem. However, contrary to NP-complete problems, NP-hard problems do not necessarily lie in NP. Up next we quote [A], that is the standard textbook on approximation algorithms.
>
> P-459: “We conclude this section by defining the term NP-hard, which can be applied to either decision or optimization problems. Roughly speaking, it means as hard as the hardest problem in NP ”
>
> P-459: Definition B.9 (NP-hard): A problem A is NP-hard if there exists a polynomial-time algorithm for an NP-complete problem B when the algorithm has oracle access to A (notice that is A not required to belong in NP.)
>
> Maximum Independent Set is not only NP-hard but in fact it is highly inapproximable. More precisely, For any $\epsilon>0$ there is no $1/n^{1-\epsilon}$−approximation polynomial-time algorithm for Maximum Independent Set unless P=NP [A-C]. As a result, solving MIS (even approximating it) is a computationally intractable task. Under this perspective, we do not understand why the following lines are plain incorrect.
>
> Line 181 -> “Since  MIS is an NP-Hard problem, annotating such data comes with an insurmountable computational burden”.
>
> Line 21 -> “Annotating 20 such data requires the solution of a huge number of instances of the CO, hence such supervised learning approaches are computationally infeasible for NP-hard problems”.
>
> [A] The Design of Approximation algorithms, Williamson et al. 2009
>
> [B] Linear degree extractors and the inapproximability of max clique and chromatic number, Zuckerman et al. 2007
>
> [C] https://webdocs.cs.ualberta.ca/~zacharyf/courses/approx_2014/notes/nov26-675.pdf
>
> [D] Learning What to Defer for Maximum Independent Sets, Ahn et al. 2020
> ____
> > Q2: Various statements are imprecise or plain wrong
>
> There could be a misunderstanding here, so please let us paste the actual line 136:
>
> “However, there might exist a reasonable graph comparing functions that i) are efficiently computable ii) lead to near-optimal solutions”.
>
> Thus, we do not claim/prove that such a graph comparing function exists. The goal of this paper is to derive such a GNN-based graph comparing function that performs well (producing large independent sets) in graphs of interest.
>
> In Remark~4 we write that the non-convex problem of Eq 2 can be treated with a first-order method such as SGD that in practice performs well in the non-convex optimization settings arising in the context of ML. The latter is verified by our experimental evaluations indicating that the self-supervised approach improves over training. Beyond the aforementioned clarifications, we are open to suggestions from the reviewer to improve the clarity of our writing.
> ____
> > Q3: Computational complexity
>
> In the revised version we will explicitly write the worst-case computational complexity of our inference algorithm that is indeed O(n^3). In Table~1 we compare the running time of our method with respect to previous deep learning approaches as well as Gurobi and SCIOPT. Note that s/g stands for seconds/graph and was used in previous works as well, e.g. [1, 2]. We also remark that the worst-case complexity of Gurobi and SCIOPT is exponential in n since otherwise $P = NP$.
> ____
> > Q4: Missing a +1 in Theorem 1
>
> Let us first emphasize that Theorem 1 is correct (a +1 is **not** missing).
>
> Notice that |MIS(G/{v})| refers to removing the node from G and thus $v$ will not be contained in the produced solution (that is a subgraph with isolated nodes). The case where $v$ is part of the produced solution is captured in |MIS (G/N(v))| at which all neighbors of $v$ are removed and thus $v$ will become an isolated node that will never be removed in the recursive calls.
> ____
> > Q5:  Variance in estimating the experimentation
>
> The variance of the process is at most $n$ since the random variable can take value at most $n$. In our experimental evaluations we used 5 roll-outs which is enough to sufficiently reduce the variance of the process.
> ____
> > Q6: Not acceptable practice to remove the RUN-CSP baseline
>
> We respectfully disagree with the reviewer. We use the open-source code (of the authors) for RUN-CSP model. When this code is trained on RB and COLLAB datasets, it obtains a poor performance. Therefore, instead of misleading the readers with a low performance, we prefer to indicate that we obtain an unexpected performance, such that other researchers can validate or refute our claim. If the reviewer has any recommendation on the topic, we will gladly change the phrasing to better reflect our case.
> ____
> > Q7: Table 1 uses an unexplained unit of s/g
>
> Seconds/graph; we will clarify this. This is standard notation in deep learning papers, such as [1].
> ____
> > Q8: Justification for benchmark datasets
>
> In our experimental evaluations we used the benchmarks used in the previous DNN approaches, e.g. [1], so as to provide a fair comparison.
> ____
> > Q9: Comparison with classical optimization solvers
>
> We remark that our method achieves the best performance across previous DNN approaches for MIS that have already been published in top-tier conferences as NeurIPS. We agree with the reviewer that learning to optimize is at an infantry level and in most CO settings of interest is outperformed with standard solvers and algorithms that have been tough developed for decades. We believe that the latter does not lower either the value of our work or the value of learning to optimize line of research.
>
> We are happy to address any other concerns the reviewer qbTV might have.
>
> [1] Karalias, Loukas. Erdos goes neural: an unsupervised learning framework for combinatorial optimization on graphs. NeurIPS, 2020

---

> > ### Author Response · Authors · 2023-08-16
> > **Are there any remaining concerns?**
> >
> > Dear reviewer qbTV,
> >
> > we are thankful for your time and effort to review our work.
> >
> > Based on your original review, our responses clarify that MIS is an NP-hard problem (correctly stated in the manuscript) and the validity of our theorem. We also elaborate on the statements flagged by the reviewer. We are wondering whether the reviewer has any other concerns regarding our work.
> >
> > Best regards,
> >
> > Authors

---

> ### Author Response · Authors · 2023-08-19
> **Thank you for the updates and the increased score**
>
> Dear Reviewer qbTV,
>
> Thank you for the revised remarks and for increasing your score.
>
> We confirm that we will do our best to improve the paper in the final version and include the recommendations of all reviewers. Concretely, based on your response:
>
>
> *Q8*. If the reviewer has any concrete datasets, we would be happy to take a look and try to conduct a preliminary experiment in the final version. That being said, the datasets that we used are the ones that previous DNN approaches use for MIS, with those papers being accepted in top-tier machine learning conferences, often as orals [A].
>
>
> *Q2*. We will rephrase Remark 4.1 to avoid any misunderstanding.
>
> *Q5*. We will explicitly mention the number of roll-outs used to estimate the expected value and the variance.
>
> *Q6*. We will further clarify that the RUN-CSP presented unexpectedly poor performance.
>
> *Q7*. We will explicitly mention graphs/second. We are thankful for bringing this to our attention.
>
> *Q9*. We will further discuss the limitations of our approach and we sincerely hope that the research community further builds on our result and develops the interplay between Dynamic Programming and self-training technique
>
> Once again, we are thankful for the revised remark and the revised score.
>
> [A] Karalias, Loukas. Erdos goes neural: an unsupervised learning framework for combinatorial optimization on graphs. NeurIPS, 2020.

---

### Official Review · Reviewer_rVy4 · 2023-07-26

**Soundness:** 3 good
**Presentation:** 2 fair
**Contribution:** 3 good
**Rating:** 6
**Confidence:** 3

**Summary:**

This paper proposes a novel graph neural network (GNN) framework for solving the maximum independent set (MIS) problem. The idea is to use a randomized divide-and-conquer algorithm (which is termed as dynamic programming in the paper though) to pick a random node or its neighborhood and divide the problem into two subproblems recursively. In order to choose one of two actions at each step, the authors propose to parameterize the comparator with a GNN over the current graph structure. The advantage of this method is that it can bootstrap training data generation with a trained GNN, and then improve the GNN with the new training data. This reduces the cost of training data generation for MIS, which is an NP-hard problem. Experiments on 4 datasets show that the proposed CMP method achieves the best results among all deep-learning-based methods under similar time budgets.

**Strengths:**

- S1: This paper proposes a novel solution for the MIS problem. It induces a random MIS inference algorithm from a comparator function, where the comparator function is parameterized by a GNN model.
- S2: One of the challenge for NP-hard problems is the cost for generating annotated data. The authors employ a smart way to bootstrap data generation from a learned GNN comparator and use the generated data to further train the model.
- S3: Experiments on 4 datasets show that the proposed CMP method achieves the best result among all deep-learning-based methods under similar time budgets.

**Weaknesses:**

- W1: The paper mistakes recursive algorithms for dynamic programming. This makes the paper somehow hard to understand. Dynamic programming is for problems with overlapping sub-problems. If the sub-problems are non-overlapping, then it is called divide-and-conquer algorithms. If every action only results in a single sub-problem, then it is merely a normal recursive algorithm. According to Alg. 1, the proposed model is a recursive algorithm.
- W2: The writing of this paper can be improved. The authors don’t emphasize the randomized nature of their inference algorithm very much, so it’s hard to understand what an expectation over the inference algorithm is at first glance. As the randomized inference algorithm is the key technique to achieve the goal of self-training, the authors may emphasize it more in the abstract and the introduction.
- W3: There are some concerns regarding the time complexity of the proposed CMP model. According to Alg. 1 and Eqn. 1, the time complexity of this paper is $O(n^2|MIS(G)|)$ in the worst case. The datasets used in this paper mostly contain around a hundred nodes, which can’t reveal the disadvantage in the time complexity. It’s recommended that the authors analyze this in the paper. Also is it possible to avoid $O(n^2)$ dense propagation in the GNN?

**Questions:**

- Q1: Line 20-21: According to [1], any polynomial-time sampler generates an easier sub-problem of an NP-hard problem. It looks to me that the CMP model can’t avoid this problem, either. Can you explain that?
- Q2: Line 22-23: How do you circumvent this difficulty? Please be explicit in the introduction.
- Q3: Algorithm 1: Please emphasize this is a randomized algorithm.
- Q4: Line 183-184: This sentence requires more clarification. From my understanding, a model can’t learn anything from what it infers, unless it is a randomized algorithm or some planning algorithms as in this paper.
- Q5: Algorithm 2 Line 5: Where is $G_{init}$ used?
- Q6: Line 240-241: It’s a little bit confusing that you replace the basic pipeline, which is something you proposed, with two other approaches. You may say “improve the basic pipeline by …”. The title of the paragraph may be changed to “Better MIS estimation”.
- Q7: Line 249: Mixed roll-out variant should be parallel with the first approach, not with “Estimating the MIS”. It may be renamed to “Mixed roll-out estimator”.
- Q8: Line 316-317: What do you mean by “core modules”?

[1] Yehuda et al. It’s not what machines can learn, it’s what we cannot teach. ICML 2020.

**Limitations:**

The authors discuss the limitations but not the societal impact in the paper. It seems that the proposed method will not cause obvious negative societal impact, but the authors are encouraged to discuss it.

---

> ### Author Rebuttal · Authors · 2023-08-09
>
> We thank the reviewer rVy4 for their thoughtful feedback. We address their concerns below:
>
>
> > W1: Is dynamic programming appropriate for this paper?
>
> MIS is a problem with overlapping subproblems - given the maximum independent set of the sub-graphs $G/\{u\}$ and $G/\{N(u)\}$, one can directly find the maximum independent set of G. Notice that the computing MIS on $G/\{u\}$ and $G/\{N(u)\}$ are overlapping subproblems since $G/\{u\}$ and $G/\{N(u)\}$ share both edges and nodes. That is the reason we used the term Dynamic Programming. That being said, we are open to suggestions for improving the clarity of our work.
>
> ____
>
>
> > W2: Randomized nature of the inference algorithm
>
>
> Thank you for the remark. The randomized version of our inference comes from the fact that at each level of the recursion, the under examination node is sampled uniformly at random. In our revision we will further emphasize the randomized nature of our approach as well as its importance.
>
> ____
>
> > W3: “There are some concerns regarding the time complexity of the proposed CMP model. According to Alg. 1 and Eqn. 1, the time complexity of this paper is  in the worst case. The datasets used in this paper mostly contain around a hundred nodes, which can’t reveal the disadvantage in the time complexity. It’s recommended that the authors analyze this in the paper.”
>
> The reviewer is right, the time complexity of our algorithm is $O(n^3)$ in the worst case. In the revised version of our work we discuss the time-complexity of our method in detail. In the experimental evaluations we used the exact same datasets of the previous DNN approaches for MIS at which our method performs similar or even better performance with respect to the running time (see s/g in Table 1). The main scope of this work is how to blend ideas from theoretical computer science with neural network approaches. Concretely, we aim to investigate how to couple classical algorithmic design techniques with deep learning models. Scaling our method as well as the previous DNN approaches to graphs of millions of nodes is an interesting and important future research direction. We will discuss this in the final version.
> ____
> > W4:  Is it possible to avoid  dense propagation in the GNN?
>
> We appreciate the recommendation from the reviewer. Inspired by this, we conducted the related experiment and verified that the complexity without the anti-neighbors is lower, but the performance deteriorates. The results in Table 1 (in the 1-page pdf) indicate how the performance without the anti-neighbors (AN) deteriorates. We will include the experiment in the revised version.
> ____
>
>
> ### Questions
>
> Q1: We first remark that using data sampled from a distribution of “easier instances” with respect to the worst-case instances of the CO setting of interest, is not necessarily futile. In fact the key motivation of the learning to optimize literature is that in many cases, the instances of interest admit an easier combinatorial structure. Secondly, our approach does not lie in the considered setting of [1] since it uses a polynomial-time sampler that is not static but evolves over time as the weights of the GNN model evolve (notice that our dataset is augmented according to the choices of the model at each epoch).
>
> Q2: We circumvent the difficulty of annotating such data with our DP/bootstrapping approach of self-annotation through our GNN-induced algorithm. In the revised version of our work we will further elaborate on this part.
>
> Q3-Q4: We will fix those, thank you for the suggestion.
>
> Q5: $G_init$ is just a randomly sampled graph from the distribution $\mathcal{D}$.
>
> Q6-Q7: Thanks for the suggestion, we will update to “Better MIS estimation” and “Mixed roll-out estimator”.
>
> Q8: We appreciate the attentive reading of the reviewer; we meant that we used only basic modules and not more advanced architectures, e.g. GAT, etc. We will rephrase it to `basic components’ to avoid any confusion.
>
> We are happy to address any other concerns the reviewer rVy4 might have.

---

> > ### Author Response · Authors · 2023-08-16
> > **Are the concerns of the reviewer rVy4 addressed?**
> >
> > Dear reviewer rVy4,
> >
> > we are thankful for the constructive questioning and the overall appreciation of our work.
> >
> > Our responses clarify the why dynamic programming is appropriate, the randomised nature of the inference as well as the questions raised by the reviewer.  We are wondering whether the reviewer has any remaining questions. We are happy to elaborate further, since we consider this work introduces novel elements, e.g., with respect to data annotation in MIS.
> >
> > Best regards,
> >
> > Authors

---

> > ### Comment · Reviewer_rVy4 · 2023-08-20
> >
> > Thanks the authors for their detailed reponse. The authors successfully addressed my concerns. As my score is already positive, I tend to keep my score.
> >
> > Regarding W1, I guess what you want to express is: although $G/u$ and $G/N(u)$ are not the same problem, their subproblems may share and it constitutes dynamic programming. Is that right?

---

> > > ### Author Response · Authors · 2023-08-21
> > > **Thank you for the support and the strong feedback**
> > >
> > > Dear Reviewer rVy4,
> > >
> > > Thank you for your response. We are glad that our response addressed your concerns.
> > >
> > > *Regarding W1,... Is that right?:* Yes, this is exactly what we mean. In the final version of the paper we will include the discussion on the latter.
> > >
> > > If the reviewer has any other suggestions, they are more than welcome.
> > >
> > > Thank you again for your support and constructive feedback.
> > >
> > > Best regards,
> > >
> > > Authors

---

### Author Rebuttal · Authors · 2023-08-09

Dear reviewers,

We are thankful for your time and effort to handle the paper. Please find enclosed the single-page pdf. We respond to each question of the reviewers below.

---

### Decision · Program_Chairs · 2023-09-21

**Decision:**

Accept (poster)

**Comment:**

The paper has received reviews (after rebuttal) that consensually vote for acceptance. Reviewers state that the presented approach is nice and novel with solid experimental results. The AC follows this recommendation.